# Evaluation of the Synergies of Land Use Changes and the Quality of Ecosystem Services in the Andean Zone of Central Ecuador

Yadira Carmen Pazmiño [1], José Juan de Felipe [1], Marc Vallbé [1], Franklin Cargua [2] and Yomara Pazmiño [3,*]

1   Department of Mining, Industrial and ICT Engineering, Manresa School of Engineering, Universitat Politècnica de Catalunya, EPSEM-UPC, Av. Bases de Manresa 61-73, 08242 Manresa, Spain; yadira.pazmino@upc.edu (Y.C.P.); jose.juan.de.felipe@upc.edu (J.J.d.F.); marc.vallbe@upc.edu (M.V.)
2   Research and Development Group for the Environment and Climate Change, Escuela Superior Politécnica de Chimborazo, Riobamba 060150, Ecuador; franklinccargua@gmail.com
3   Instituto Superior Edwards Deming, Quito 170904, Ecuador
*   Correspondence: ypazmino@deming.edu.ec

**Abstract:** The scarcity of information that allows for understanding the importance of natural resources from an economic approach is often a limitation to establishing parameters related to environmental investment in conservation plans. This study proposes a methodology that allows for modeling the variability of páramo land uses and the EV of the Chambo-Ecuador sub-basin from bioeconomic monitoring that links the economic rent of páramo land uses with remote sensing tools and geographic information systems. Multilayer Perception, Markov Chains, and Automata Cells algorithms were efficient for the detection of land uses in páramo; the normalized differential humidity index was the most relevant variable to identify crops, showing that leaf properties and water stress are linked to crop yields in the Andean region. The páramo decreased by 13% between 2000 and 2010, increasing its degradation to 19% between 2010 and 2020. A 28% reduction is expected between 2000 and 2030; the EV between 2000 and 2020 was $2.86 \times 10^8$ and $2.59 \times 10^8$ respectively. In 2030, EV is expected to decrease to $2.48 \times 10^8$. Transitions in land use and EV are associated with productive dynamics, which decrease environmental services, such as water retention and carbon storage, intensifying changes in the ecosystem climate.

**Keywords:** páramo; conservation; ecosystem value; soil; conservation; environment

## 1. Introduction

The páramos are high mountain ecosystems of volcanic origin, located between 3500 and 5000 m.a.s.l, and extend from the Andean Mountain range of Mérida in Venezuela, through Colombia and Ecuador to northern Peru. Ecuadorian páramo ecosystems cover an area of 1,250,000 ha and extend from the northern border with Colombia to the southern border with Peru, through the Andean corridor [1]; the Andean zones harbor a large number of alpine ecosystems, with about 628 varieties of endemic plant species, equivalent to 15% of the vegetation of the area and 4% of the total vegetation of the Ecuadorian territory [2].

The environmental services (ES) of the páramos are fundamental for the development and equilibrium of their environment and are characterized by their high capacity for carbon sequestration, water regulation, and high levels of biodiversity [3]. Runoff from páramos is estimated to be between 506 and 933 mm of water per year. This value is equivalent to 2/3 of the annual precipitation of the Altiplano region. The páramo can retain up to 400 t/ha of carbon [4].

Population growth in the páramo zones is accompanied by economic activities that cause changes in the soils; the primary economic activities in the study area are based on agriculture and livestock, while the secondary activities are based on mining and forestry. Most of the settlers engaged in agriculture plant crops without any planning system that

included fallow periods and soil spatial logistics [5]. Cattle ranchers burn Andean vegetation to fertilize the soil to increase pasture productivity [6]. At first glance, silvicultural activities favor ecosystem stability by planting trees in the páramo; however, this is not true, since tree planting eliminates native vegetation in the area, causing alterations in the ecosystem [7]. The environmental impacts caused by economic activities are associated with water availability and soil fertility. Studies of the causes that contribute to the degradation of ecosystem valuation (EV) related to páramo soils are key to identifying the moment when the ecosystem will move from a stable to a critical state [7,8].

In relation to land use assessment, several studies recommend the use of remote sensing data and geographic information systems (GIS) to monitor areas that are difficult to access [9,10]. According to Delalay et al. [11] significant progress has been made in satellite image classification due to the emergence of machine learning methodologies. There is a large body of research focused on the study of land cover classification techniques from satellite information [12].

Several studies have evaluated land use transitions using multi-criteria analysis (MCA) to assign weights to various factors based on input from decision-makers. This method facilitates the evaluation of the relative effects of each criterion in land use modeling. However, the subjectivity of the input values for the weights represents a limitation of the model [13]. Other studies have used methods such as k-means and support vector machines. The k-means method uses MODIS data that have a high temporal observation frequency but limited geographic resolution. This methodology is suitable for the analysis of natural disasters due to its short tracking interval. Support vector machines (SVMs), base their technique on supervised learning classification and identify the best hyperplane to achieve a clear separation between a dataset. Unfortunately, the practical use of SVMs is limited by the quality of their hyperparameter settings, which have a direct impact on their performance [14].

The maximum likelihood algorithm, which is also used in soil analysis, assumes that the data follow a normal distribution function to assign the probability that any pixel belongs to each of the classes. However, the hypothesis that reflectivity data follow a normal distribution is not always satisfied and must always be verified [15]. Algorithms integrated by artificial neural networks have recently become an important focus of research, as they represent a robust approach to modeling complex land use behaviors. Unlike the multivariate approach, they do not require assumptions about spatial autocorrelation and multicollinearity of the data [16].

In relation to ecosystem valuation of páramo land uses, different methodologies based on monetary terms and remote sensing have been used. For example. In southeastern Ecuador, a local analysis of the perception of páramo EV was carried out using the avoided cost method and spatial data. The authors determined that the land uses that caused the greatest impact on the area's VE were agriculture and livestock and that the VE from 2010 to 2020 decreased by 6%. The limitation of the avoided cost methodology is that it uses dose-response functions, which require varied data for valuation, which may not be available to the researcher [17].

In another study conducted in San Juan Chimborazo (Ecuador), the EV of Andean soils was studied using techniques based on remote sensing, analysis of physical and chemical parameters, and market and administrative values. The monetary weights used in the study were obtained by applying the theories of opportunity cost and benefit transfer methodologies [18]. The EV results of the study are general since the economic indices used were not processed by any adjustment method, to represent the reality of the area. In the northern and southern departments of Santander, in Colombia, the EV of the páramo was determined using travel cost, benefit transfer, and satellite information methodologies. The authors described those environmental services (ES) from soil provision have a direct impact on cultural ES [19]. The limitation of the travel cost methodology is that it only allows valuing impacts on local public goods based on ex-post situations.

Outside South America, in Buddhashanti (Nepal) and northwestern Yunnan (China), several authors have calculated the EV of soils using integrated valuation of ecosystem services and compensation (InVEST) models and remote sensing. Their work described that the loss of EV is mainly linked to population expansion and the agrocultural frontier and that the most affected areas are those located in latitudes accessible to human development [20]. The drawback of the compensation method for calculating EV is that it is based on the perception of the maximum willingness to pay by ES beneficiaries and often this perception does not reflect reality. To date, most of the studies analyzing VA related to páramo land use in South America have been conducted in Colombia. Few relevant ecosystem studies exist in Ecuador, most of which cover the northern zone, followed by the southern zone [21]. Furthermore, the few existing studies use methodologies that are not extrapolable, as they are based on local factors.

For the study of land use in the páramo, this work used an artificial multilayer perception (MLP) artificial neural network, which has gained recognition as a highly reliable neural network model. MLP has demonstrated its effectiveness in land cover classification and pattern recognition tasks [22,23]. The neural network model consists of an input layer, hidden layers, and an output. MLP uses spectral and spatial information from satellite images as input information to the network [24]. The model can learn to associate these input features with land use classes through the learning process. Although it is a reliable prediction model, its performance is often impaired by errors in the adjustment of variable weights by the modeler [25]. This work will strive to reduce this type of error as much as possible by adjusting the weights through a comparison of the training values, the network output values, and the expected values. The error will be reduced by comparing the interactions of the weights on the output signals, through the feedback of the network, based on the adjustment of the functions used in the synaptic links of the neurons.

To strengthen the analysis of the ecosystem value of the páramo in economic terms, this study employed the opportunity cost-benefit transfer methodology following the pioneering work of Costanza et al. [26], who estimated coefficients to determine the EV of natural biomes. The present work focuses on adjusting the coefficients determined by Costanza to make them applicable to the Andean areas of Ecuador, using economic indices.

Based on the above, the objectives of this research are the following: (a) evaluate the less studied area of the Ecuadorian páramo, through bioeconomic monitoring, remote sensing techniques, geographic information systems, and market values to determine its conservation status; (b) optimize the performance of the artificial neural network of multilayer perception (MLP) and adjust the weights of the variables in the synaptic links of the algorithm to improve the classification of land uses in the study area; (c) determine coefficients through economic indexes to analyze the ecosystem valuation of the environmental services of land uses in páramo.

The findings of this study will contribute to establishing payment strategies for ecosystem services as economic instruments to help adopt land use practices that guarantee the provision of ecosystem services to the community with a vision of the future and open spaces to understand the importance of the páramo in environmental and economic terms.

## 2. Materials and Methods

### 2.1. Study Area

This study focuses on the páramo zone of the Chambo sub-basin, located in central Ecuador (Figure 1), the sub-basin is made up of seven cantons: Alausí, Colta, Guamote, Guano, Penipe, Riobamba, Chambo (Figure 2). It has an area of approximately 3580 km$^2$, of which 42% is páramo and covers 51% of the total area of the province. The páramo area in the province represents 17% of the total páramo area in the country. There are two protected areas in the province, the Chimborazo Fauna Production Reserve and Sangay National Park. The páramo ecosystems are located in an altitudinal band between 3000 and 4500 m.a.s.l. and contain almost 30% of the vascular plant species, which demonstrates the great representativeness of this ecosystem [27].

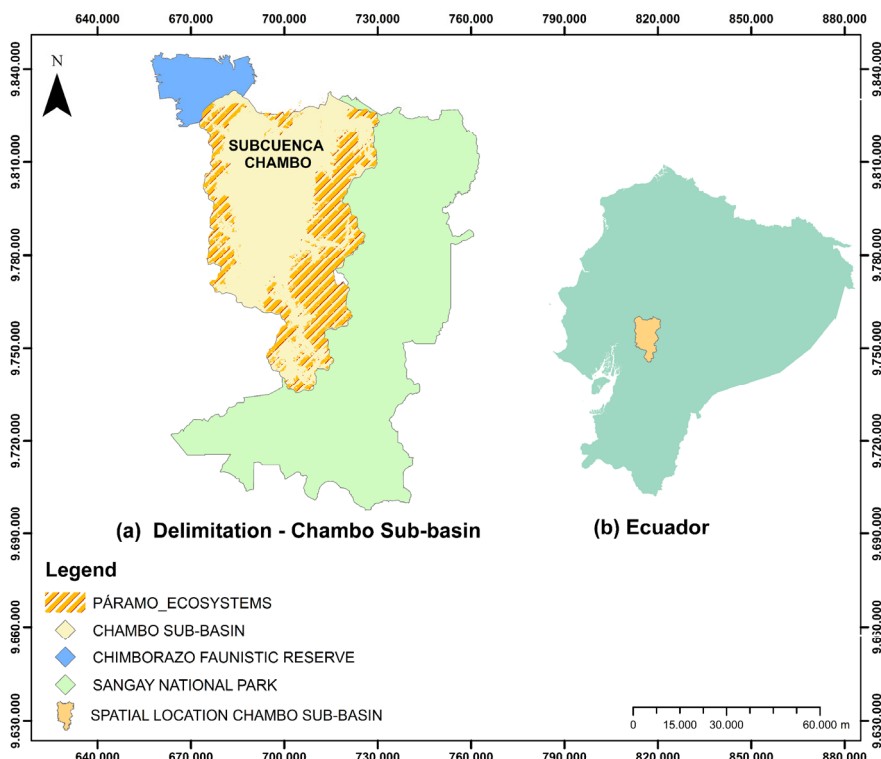

**Figure 1.** Location of the study area.

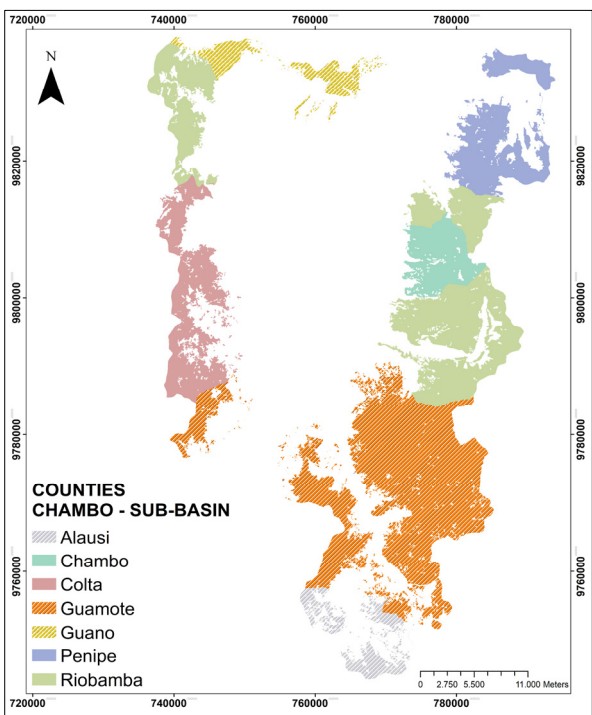

**Figure 2.** Counties Chambo sub-basin.

### 2.2. Workflow

The research was carried out in the following phases (Figure 3): determination of land use, land use changes, and ecosystem valuation.

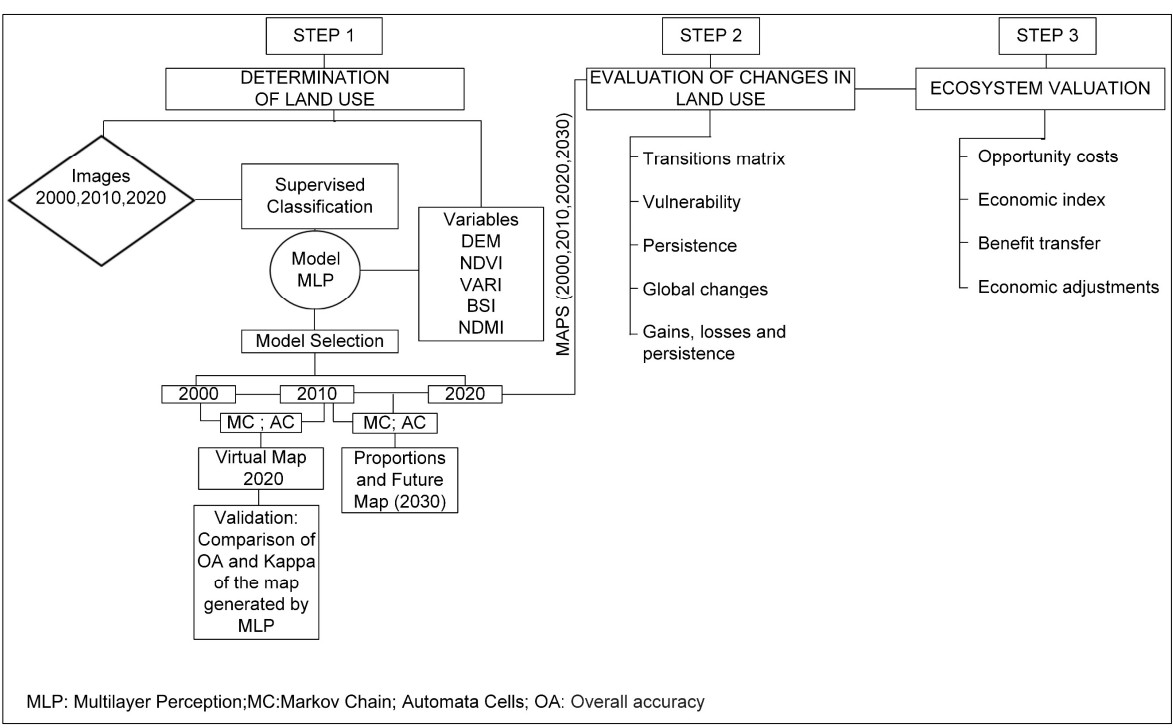

**Figure 3.** Flowchart: Main stages of the study.

*2.3. Determination of Land Use*

2.3.1. Image Processing

The images used for land use classification were downloaded from NASA's official web page https://earthexplorer.usgs.gov, accessed on 3 February 2023. From Landsat 7 and Landsat 8 satellites, considering a cloud cover of less than 30% so that the visible field is as wide as possible. Twenty satellite scenes from the years 2000, 2010, and 2020 with path = 10 and row = 61 were used for the central area of the country and aggregated into a single layer. The atmospheric correction and geometric verification procedures of the images were performed with QGIS 3.4 software. The software was downloaded on January 20, 2023 through the following page https://qgis.org/es/site/forusers/download.html [28]. Atmospheric correction was performed using the dark object subtraction method, which aims to correct the electromagnetic energy scattering effect of water particles suspended in the atmosphere [28]. The geometric accuracy of the images was verified using topographic maps and base maps of rivers and roads, georeferenced in the UTM Datum WGS84 projection of the Ecuadorian Military Geographic Institute [29].

2.3.2. Checkpoints

For the classification of land use for the years 2000, 2010, and 2020 in the study area, 3580 control points were established, 70% of which were verified in situ. These years were selected because they represent the years with the greatest growth of the main activities related to the degradation of the páramo, in addition to the fact that in recent decades the increase of anthropogenic activities on the natural systems of the area has increased [30,31]. Land uses were established based on information from the land cover inventory of natural ecosystems mapping developed by the Ministry of the Environment of Ecuador [27]. The land uses included were Crops (C), Pastures (Pz), Páramo (Pr), Forest Plantations (PF), and bare soil (S).

2.3.3. Variables

The following variables were used to analyze the land cover of the páramo in the study area: Normalized Difference Vegetation Index (NDVI), height (DEM), Green At-

mosphere Resistant Visible Vegetation Index (VARI), Bare Soil Index (BSI), Normalized Difference Moisture Index (NDMI), and Soil Organic Carbon (SOC). The digital elevation model (DEM) [29] was obtained from the data sources of the Military Geographic Institute of Ecuador; the GSOC, which was taken from the Global Soil Organic Carbon Map databases [32], and the spectral values were determined through Python of QGIS 3.4 by means of the association of bands determined by the QGIS 3.4 software Programming Guide. Table 1 details the formulas used for the calculation of the spectral indices.

**Table 1.** Spectral values were obtained from Landsat 7 and Landsat 8 bands.

| Index | Formula | Characteristics |
| --- | --- | --- |
| NDVI: Normalized difference vegetation index | $\text{NDVI} = \frac{(NIR-RED)}{(NIR+RED)}$ | The normalized difference vegetation index (NDVI) was used to separate the vegetation from the brightness produced by the soil and to relate photosynthetic activity and leaf structure of the plants, allowing to determine the vigorousness of the plants [33]. |
| VARI: Visible vegetation index | $\text{VARI} = \frac{(GREEN-RED)}{(GREEN+RED-BLUE)}$ | The visible vegetation index (VARI) allowed to observe vegetation in the visible section of the spectrum, mitigating both illumination divergences and atmospheric factors [9]. |
| BSI: Bare soil index | $\text{BSI} = \frac{[(SWIR+RED)-(NIR+BLUE)]}{[(SWIR+RED)+(NIR+BLUE)]}$ | It helps to identify areas of soil without vegetation, soils with vegetation and soils with scarce vegetation [34]. |
| NDMI: Normalized difference moisture index | $NDMI = \frac{(NIR-SWRI)}{(NIR+SWRI)}$ | The Normalized Difference Moisture Index (NDMI) contributed to the detection of moisture levels in vegetation using a combination of NIR and SWIR spectral bands. It is an indicator of water stress in crops [35]. |

Where: *NIR:* near infrared (landsat 7: band 4; landsat 8: band 5); *RED:* (landsat 7: band 3; landsat 8: band 4); *BLUE*: (landsat 7: band 1; landsat 8: band 2); *SWRI*: band 7; *GREEN:* (landsat 7: band 2; landsat 8: band 3).

### 2.3.4. Land Use Classification Algorithms

The information obtained from the variables was processed using the Multi-Layer Perception (MLP) modeler. MLP method is a supervised learning algorithm that is developed through training using a known data set. MLP allowed modeling and relating variables to generate a logistic regression and a powerful neural network for the development of sub-models of the studied coverages [36].

The product in raster format for the years 2000, 2010, and 2020 and the data source developed according to the variables used in the research were processed in TerrSet v.19.0.8 software. The software was downloaded on 20 January 2023 through the following website https://clarklabs.org/download/terrset-2020-service-update/ [37] to develop the processes of the MLP method. TerrSet is an integrated geospatial system that allows for monitoring and modeling the earth system for sustainable development [38].

The pixel samples of the different equal-sized coverages that changed over time and the pixels that persisted between the two dates were used as training data in the dynamic learning process of the MLP [36]. Seventy percent of the values were used during the training stage and 30% in the validation stage.

Figures 4 and 5 detail the transmission processes executed to obtain the values of the different layers of the MLP modeler. The layers are associated with a weighting weight that regulates the strength of interconnection between the variables. The neural units of the hidden layers receive as input signal the sum of the signals of the values introduced in the system and the output layers receive the sum of the signals of the values of the hidden layers [39].

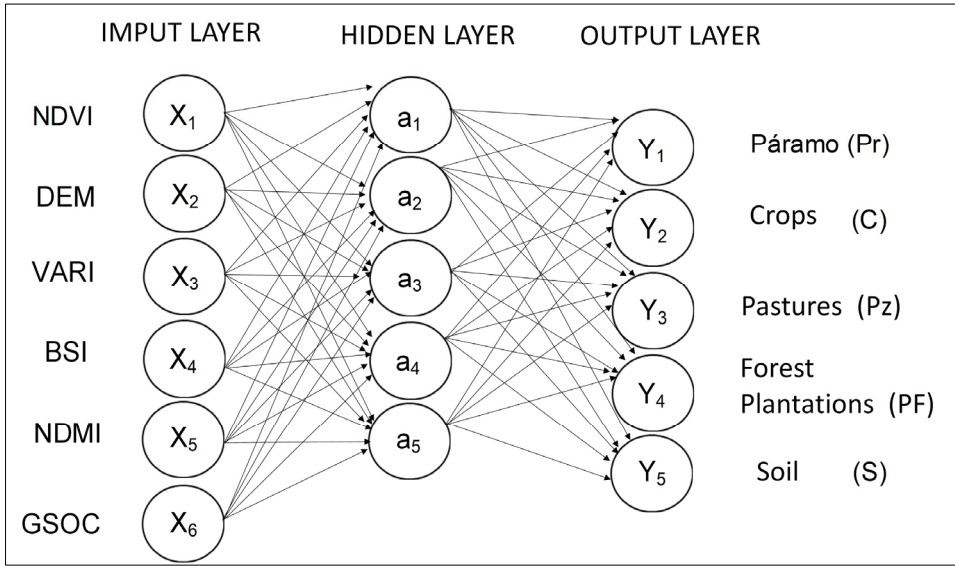

**Figure 4.** General diagram of the MLP modeler.

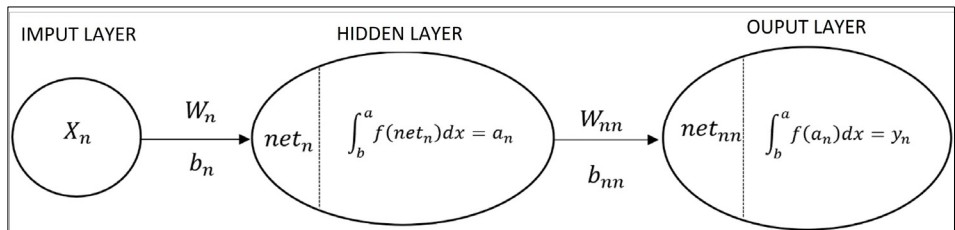

**Figure 5.** Fragment of the general scheme of the MLP modeler.

The architecture of the model is based on a forward feed network which is composed of three layers. The first layer or input layer is composed of the variables NDVI, DEM, VARI, BSI, NDMI, and SOC. The third or output layer is made up of five units representing the categories studied and the second or hidden layer houses the model activation processes.

To calculate the value of the hidden layer, we multiplied the weights by the input values and added a weight associated with a dummy neuron $b$ whose value was 1, as exemplified in Equation (1), to complete the process the value was activated by means of the rectified linear unit activation function ReLU, Equation (2) [38].

$$net_n = \sum w_n * x_n * b_n \tag{1}$$

where $net_n$ is weights and values of variables entered the hidden layer neuron; $x_n$ is value of variables used in the model; $w_n$ is weight of variables in the algorithm; $b_n$ is dummy neuron.

$$\int \left( a_{n)} = \max(x, 0) \right. \tag{2}$$

where $= x$ will take in the value of $net_n$ to identify the value of the intermediate layer.

The output layer product described in Equation (3) was obtained from the values found in the hidden layer, its weights, and fictitious weights associated to the units. As in the hidden layer the final value was obtained through the ReLU function (2) [36].

$$net_{nn} = \sum w_{nn} * a_n * b_{nn} \tag{3}$$

where $= net_{nn}$ is weights and final values of the hidden layer; $a_n$ is value of the hidden layer; $w_{nn}$ is weight of the variables in the algorithm; $b_{nn}$ is dummy neuron.

The rectified linear unit function ReLU was used in the activation process because of its advantages in convergence speed in addition to efficiently mitigating gradient fading. The function has no saturation regions so the results obtained can be more accurate [39]. To reduce the model error, the backpropagation algorithm was used, which is a gradient calculation method used in supervised learning algorithms to train artificial neural networks. Once a pattern has been applied to the input of the network as a stimulus, it is propagated from the first layer through the following layers of the network until an output is generated. The output signal is compared with the desired output and an error signal is calculated, which is backpropagated to the intermediate neuron layers, dividing it between them, this procedure is propagated backwards through all the intermediate neuron layers [36].

The weights w used in the Multilayer Perception (MLP) modeler were randomly selected from 0 to 1. The model was fitted through the backpropagation algorithm. Training values, network output values, and expected values were compared. The error was decreased through a comparison of weight interactions in the output signals by means of network feedback [38].

To carry out the backpropagation algorithm, the loss gradient variables were related by means of a computational graph (Figure 6), which allowed establishing the relationship between the nodes of the input information, random weights, activation function, response values, and error in the network [39].

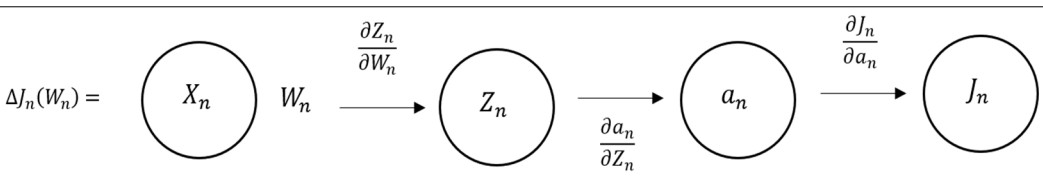

**Figure 6.** Computational graph of the loss gradient. Where = $X_n$ is network input value; $W_n$ is random weight; $Z_n$ is activation function; $a_n$ is output value of the layer; $J_n$ is error.

The gradient of the loss function was calculated by the product of the partial derivalries of the computational plot, using the following Equations (4)–(9) [36].

$$\Delta J_n(W_n) = \frac{\partial J_n}{\partial W_n} = -(d - a_n) * \frac{1}{w_n x_n} * x_n \tag{4}$$

where: $X_n$ is value of the network entry; $W_n$ is random weight; $a_n$ is output value of the layer; $J_n$ is error; $d$ is corrected layer output value.

$$\Delta J_n(W_n) = \frac{\partial J_n}{\partial W_n} = \underbrace{\frac{\partial J_n}{\partial a_n}}_{\text{Term 1}} * \underbrace{\frac{\partial a_n}{\partial z_n}}_{\text{Term 2}} * \underbrace{\frac{\partial z_n}{\partial w_n}}_{\text{Term 3}} \tag{5}$$

Term 1 of Equation (5) was determined from the mean square error function.

$$\frac{\partial J_n}{\partial a_n} = -(d - a_n) \tag{6}$$

Term 2 of Equation (5) was calculated considering that the ReLU function will take values equal to or greater than 0.

$$\frac{\partial a_n}{\partial z_n} = \frac{1}{w_n x_n} \tag{7}$$

Term 3 of Equation (5) was established considering that the ReLU function was determined in relation to the activation value resulting from the product of the network input values and the random weights [36].

$$\frac{\partial z_n}{\partial w_n} = x_n \tag{8}$$

To complete the backpropagation algorithm, the adjusted weights were calculated as follows [38].

$$w_{ajustado} = w_n - n\Delta J_n(W_n)$$
$$w_{ajustado} = w_n + (d - a_n) * \frac{1}{w_n}$$

(9)

where: $W_n$ is random weight; $a_n$ is output value of the layer; $J_n$ is error; $d$ is corrected layer output value; $d$ is corrected layer output value.

The process was repeated until the error was acceptable for each of the learned patterns.

### 2.3.5. Future Land Use Prediction Model

Based on the products generated in the MLP modeler, an analysis was carried out using Markov chains (MC) and automata cells (AC) to project land use changes for the year 2030, this procedure was performed using the TerrSet v.19.0.8 software. Markov chains (MC) are considered a predictive and practical method applicable in different scientific fields such as the study and prediction of land use dynamics, characterized by the fact that the probability of land cover change in a period only depends on the immediately previous state of the system [40].

The method allows for modeling a future scenario from two previous states where the value of time $t_2$ depends on the values of times $t_0$ and $t_1$. The analysis of Markov chains (MC) is based on the internal dynamics of the system, where it records an evolutionary trend of land use. The method considers that changes observed in a period tend to be repeated in a subsequent period [23]. Markov chains (MC) were calculated by the interactions of the transition matrices of the study period (Table 2) and the land use percentages of the previous time of the land use change system, as described in Equation (10) [40].

$$P_1 = P_o * P$$

(10)

where $P_1$ is future land use change; $P_o$ is initial land use condition; $P$ is average transition matrix.

**Table 2.** The statistical formulas applied to analyze the reliability of the models are described below.

| Measure | Formula | Description |
|---|---|---|
| "Producer accuracy" (PA) | $PA = \frac{D_{ij}}{R_i}$ | It is an accuracy based on references calculated from the predictions of the coverages under study, in this case: moorland, pasture, crops, forest plantations and bare soil. Based on the predictions, a percentage of correct and incorrect cover detections is established [41]. |
| "User accuracy" (UA) | $UA = \frac{D_{ij}}{C_i}$ | It allows to recognize the probability that a classified category actually represents the same category in the field [42]. |
| "Overall accuracy" (OA) | $UA = \frac{\sum D_{II}}{N}$ | Indicates the list of all reference categories that have been successfully identified [42]. |
| Kappa Index | $K = \frac{N\sum_{i=1}^n m_{i,i} - \sum_{i=1}^n (G_i C_i)}{N^2 \sum_{i=1}^n (G_i C_I)}$ | Analyzes the match between the data observed in the image and the data identified in the classification [43]. |
| Profit Degradation Persistence | Gain: $(G_{ij}) = P_{+j} - P_{jj}$ <br> Degradation: $(L_{ij}) = P_{j+} - P_{jj}$ <br> Persistence: $P_{+j} - Gain$ | It contributes to the identification of changes in terms of recovery, loss and exchanges between the categories analyzed [43]. |

$D_{ij}$: Successfully recognized pixels; $R_i$: Reference pixels of the same coverage; $C_i$: Total classified data belonging to a coverage; $N$: Total data of the failure matrix; $P_{+j}$: Gains in the second time; $P_{jj}$: No change; $P_{j+}$: Degradation in the second time; $gp$: Divergence between gains in the second time and persistences.

The transitions were determined by means of a multi-criteria evaluation of variables related to páramo land cover. Ten interactions of the calculation of future land use were carried out to project the land cover transition for the year 2030. In each interaction, the initial condition was updated considering that the result depends on the previous change to the future projection system. The páramo land use map for the year 2030 was obtained

using automata cells (AC). Automata cells (AC) are capable of modeling very complex spatial dynamics of the territories. The cell matrix visualizes the probability of land use change, depending on the proximity of each cell [23].

Once the classification criteria were defined and the páramo land use changes were hierarchized in the previous procedures of the study, weighting factors were defined to establish the relative importance of each land use with respect to the change suitability criteria. The weights were applied to each pixel according to the score of the categories in ascending order according to the capacity for change.

The 2020 land cover map was used as a reference to determine the areas of land uses evaluated. The decision line was moved from highest to lowest according to the proportion of coverage in the base map until the necessary raster cells were captured to reach the future area of change determined by the Markov procedure.

To evaluate the accuracy, a comparison was made between the actual map for the year 2020 and the prediction map for the year 2020 using the Validate option of the TerrSet v.19.0.8 software to establish the Kappa variations, allowing to recognize the similarities between the classification of the projected map and the base map for 2020.

The uncertainty of the prediction model was determined by considering data quality, model validation, and a sensitivity analysis. The values used were thoroughly reviewed to ensure that they were meaningful for the study area. Verification of the land use models was performed through the total number of control points incorporated by the operator for each of the land covers. The performance was determined by the total number of objects recognized by the software. The level of uncertainty will be obtained by analyzing the data observed in the satellite scenes and the data determined by the classifier [23,38].

Sensitivity analysis made it possible to evaluate the robustness of the Markov chain model. The analysis is based on systematically varying the input parameters or assumptions of the model and observing the resulting changes in the predictions, the sensitivity of the model to different scenarios can be evaluated. This analysis helped to identify critical factors that significantly influence model predictions and provided information on the reliability of the model under various conditions.

Through the study of data quality, model validation, and sensitivity analysis, model performance and uncertainty were established [23]. The statistical measures used in these processes are listed in Table 2.

### 2.4. Evaluation of Land Use Change

The generated maps were evaluated by means of the following statistical measures (Table 2).

### 2.5. Estimation of the Ecosystem Value of the Zone

The ecosystem valuation (EV) was carried out through the opportunity costs of agricultural use of the páramo soils and the transfer of benefits of the land covers included in the analysis. Economic values from the study "The value of ecosystem services and the world's natural capital" were used [26,44]. Although the EVs were derived from a substantial collection of previous research on natural biomes, including the páramo ecosystem, the factors for the equivalent weights of agricultural uses were not completely appropriate for the local scale of the study site. Therefore, the factors in Equations (11) and (12) were modified. The EVs for the Pr, PF and S categories were taken directly.

$$E_f = (\frac{C}{C_o})E_{of} \tag{11}$$

where = $E_f$ is ecosystem value factor; $C$ is average value of crop production per unit area of the páramo soils of the study area; $C_o$ is average value of crop production per unit area of the Andean region of the country; $E_{of}$ is factor of the equivalent weight of the EV.

The average value of crop production per unit area of the páramo soils in the study area was determined from the income, expenses, rate of return, and net profit. The income

was calculated from the relationship between the yield and the cost of the crop, the net profit was determined based on the income and investment of the crop, and the rate of return was established based on the percentage obtained by the producer for each dollar invested. The parameters and investment values were taken from the consumer price index (CPI) of Ecuador, considering the impacts of price changes and projections between different years during the study period.

The average value of crop production per unit area of the Andean region of the country was taken from the official databases of the Ministry of Agriculture and Livestock [45–47] and the equivalent weight factor was taken from the study "The value of ecosystem services and natural capital of the world" [26,48]. The pasture land use was determined based on the dairy sector. The Sierra region where the Chambo sub-basin is located, is one of the main milk producing areas, contributing 75.9% in milk production at the national level [46,48]. According to the integrated water resources planning study, the citizens of the study site have based their economic production primarily on the dairy industry as a result of the stabilization of product costs [49].

To estimate the value of pasture land uses in relation to the dairy sector, the profitability of land use was determined based on the production costs referenced by the Ministry of Agriculture and Livestock and the consumer price index (CPI) of Ecuador considering price changes and projections between different years during the study period.

$$E_f = (\frac{Gr}{Gr_o})E_{of} \tag{12}$$

where $E_f$ is ecosystem value factor; $Gr$ is average value of milk production per unit of pasture in relation to the area of páramo soils; $Gr_o$ is average value of milk production per unit of pasture in relation to the area of páramo soils in the Andean region of the country; $E_{of}$ is factor of the equivalent weight of ecosystem services.

To calculate the average value of milk production per pasture unit in relation to the area of moorland soils, the monetary value perceived by pasture land uses in relation to the dairy sector was determined for which the profitability of land use was estimated from the production costs referenced by the Ministry of Agriculture and Livestock [46].

Cattle feed consumption was analyzed based on the nutritional requirements of dairy cows. The most representative type of pasture in the Andean zones is Kikuyo (Pennisetum clandestinum), so the analysis was carried out according to the properties of this type of pasture; rotation systems and number of paddocks per hectare were established in relation to the characteristics of Kikuyo [46].

The pasture load (PL) to be consumed by each dairy cow in the Andean zone was evaluated considering the weight per animal unit (AU), this calculation is described in Equation (13). The milk production of the cows was determined using the average values of the Dairy Sector Report in Ecuador [46,50]. The AU should consume 3% of its live weight (LW) of dry matter and 0.186 additional kilograms for each kilogram of milk produced, it was considered that the pasture is made up of 80% water and 20% dry matter. The relationship between pasture load, green forage capacity per paddock and animal units made it possible to determine the feed supply capacity of the area [46].

$$PL = ((3\%LW) + (0.186Kg * kg\ de\ leche\ producido))/0.20 \tag{13}$$

where $PL$ and pasture load; $LW$ is live weight.

From the calculated factors, the EV losses and gains of land uses were determined using the following Equation (14) [45].

$$EV_i = \sum_{i=1}^{n}(A_i * VC_i) \tag{14}$$

where $EV_i$ is ecosystem value; $A_i$ is area of land uses; $VC_i$ is value of the EV coefficient.

The change of EV was determined by the difference between the values calculated in 2000, 2010, 2020 and 2030. The rate of change of ecosystem value ($EV_{cr}$) during the study period was calculated using the Equation (15) [45,49].

$$EV_{cr} = \left( \frac{EV_{t_2} - EV_{t_1}}{EV_{t_1}} \right) * 100\% \tag{15}$$

where $EV_{t_2}$ is ecosystem value time 2; $EV_{t_1}$ is ecosystem value initial time.

## 3. Results

### 3.1. Accuracy of the Páramo Land Use Classification Algorithm and Future Land Use Prediction Model

Six models (Table 3) with different sets of variables were developed from Multilayer Perception (MLP); the accuracy of each model was different in the determination of coverages. Model 1, selected because its accuracy was the highest, has all the variables which indicates that each variable makes a different contribution to the model fit.

**Table 3.** Accuracy of models.

| Model | Variables | Accuracy (%) |
|---|---|---|
| 1 | NDVI, DEM, VARI, SOC, BSI, and NDMI. | 81 |
| 2 | NDVI, VARI, SOC, BSI, and NDMI. | 71 |
| 3 | NDVI, DEM, VARI, and SOC. | 65 |
| 4 | NDVI, DEM, and VARI. | 58 |
| 5 | NDVI and DEM. | 50 |
| 6 | NDVI. | 49 |

The accuracy of model 1 was increased by entering the variables in the following order: NDVI, DEM, VARI, GSOC, BSI, and NDMI. The hierarchical ranking of the variables could be associated with their relevance in the categorization and performance of the model.

The Normalized Difference Vegetation Index (NDVI) was essential to differentiate vegetation from other land cover types and to determine its general condition, the altitude (DEM) allowed to recognize the influence of altitudinal floors on the development of land covers in the páramo, the Visible Green Atmosphere Resistant Vegetation Index (VARI) contributed to the quantitative estimation of the fraction of vegetation covers. Through the Bare Soil Index (BSI) data, it was possible to determine the areas of bare soil in places with scarce vegetation and reduce the incidence of humidity, the Normalized Difference Moisture Index (NDMI) made it possible to detect the conductivity of the land cover, and the soil organic carbon (SOC) variable was essential in the study of the correlation between carbon and the different classes of land use at different altitudes.

Table 4 shows the land use accuracy data for the years 2000, 2010, and 2020, while Table 5 shows the validation of the future prediction model for the analyzed land covers.

**Table 4.** Overall accuracy and Kappa coefficient.

| | Overall Accuracy (L)% | Overall Accuracy (V)% | Kappa Coefficient (L)% | Kappa Coefficient (V)% |
|---|---|---|---|---|
| 2000 | 81 | 79 | 81 | 79 |
| 2010 | 84 | 80 | 85 | 80 |
| 2020 | 85 | 82 | 84 | 81 |

**Table 5.** Accuracy validation of the prediction model.

|  |  | Real Map 2020 | Projected Map 2020 |
|---|---|---|---|
| Scenario 1 | Kappa (%) | 81 | 60 |
|  | OA (%) | 82 | 63 |
| Scenario 2 | Kappa (%) | 81 | 68 |
|  | OA (%) | 82 | 72 |
| Scenario 3 | Kappa (%) | 81 | 78 |
|  | OA (%) | 82 | 80 |

The evaluation of the accuracy of the classified coverages was excellent. The overall accuracy value in the validation phase was 79%, 80%, and 82% for the year 2000, 2010, and 2020 respectively and the Kappa coefficient values were 79%, 80%, and 81%.

Scenario 3 (Table 5) had the highest accuracy, its overall accuracy was 82% and 80% for the actual map and projected map, respectively, and the Kappa coefficient values were 81% for the actual map and 78% for the projected map. Based on these results, it can be recognized that the accuracy of the land use prediction model with respect to the accuracy of the actual map is very close, which indicates that the effectiveness of the model is efficient and therefore the interrelation of the variables used in the method are adequate for the study of páramo land use.

### 3.2. Distribution of Land Uses in the Study Area

Table 6 shows the results of the status of the study area. Figures 7–10 show geographically the changes in land cover. Tables 7–9 refer to the behavior of land use changes in relation to losses, gains, and persistence of the different land covers.

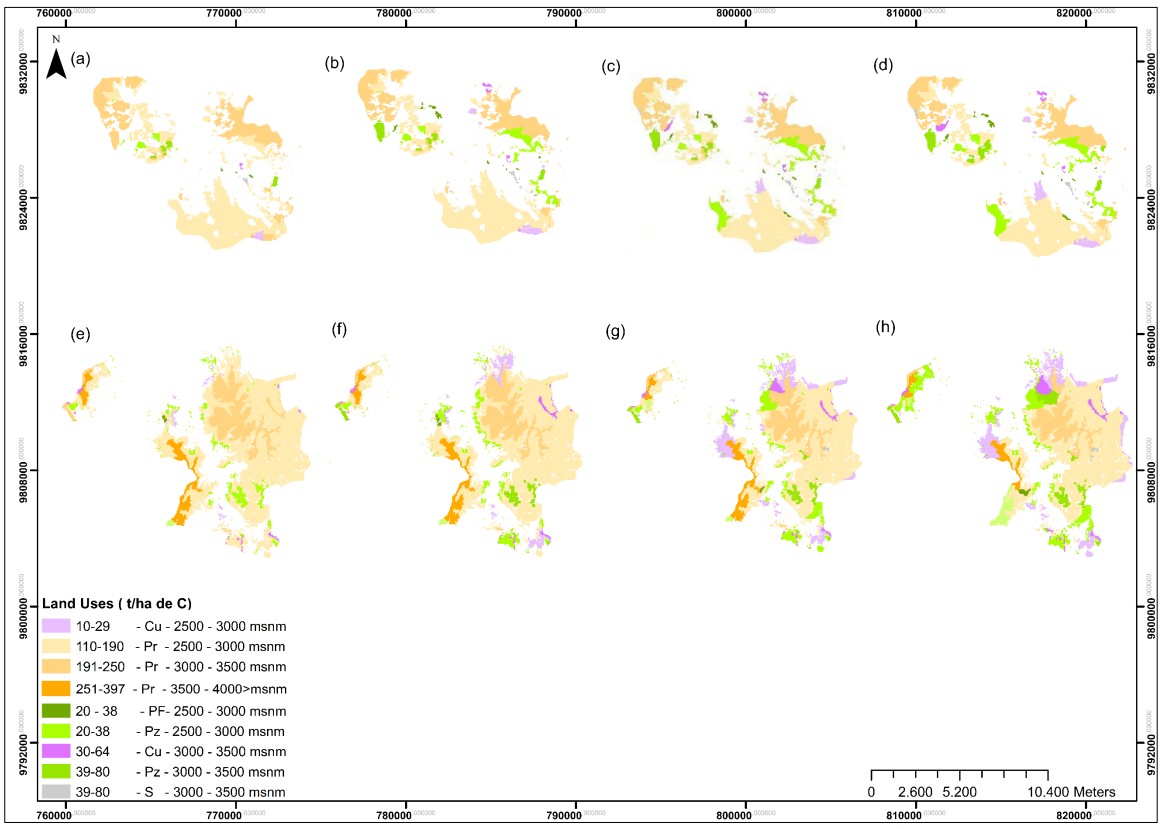

**Figure 7.** Land Uses—(**a**) Alausí 2000; (**b**) Alausí 2010; (**c**) Alausí 2020; (**d**) Alausí 2030; (**e**) Guamote 2000; (**f**) Guamote 2010; (**g**) Guamote 2020; (**h**) Guamote 2030.

**Table 6.** Land cover in the study area.

| CANTONS | COVERAGE | 2000 (%) | 2010 (%) | 2020 (%) | 2030 (%) |
|---|---|---|---|---|---|
| Alausi | Pr | 2.8 | 2.5 | 2.0 | 1.7 |
| | C | 0.0 | 0.1 | 0.3 | 1.7 |
| | Pz | 0.1 | 0.3 | 0.5 | 0.7 |
| | PF | 0.0 | 0.1 | 0.1 | 0.1 |
| | S | 0.0 | 0.0 | 0.0 | 0.1 |
| Chambo | Pr | 5.8 | 5.0 | 4.3 | 3.7 |
| | C | 0.1 | 0.3 | 0.5 | 0.6 |
| | Pz | 0.3 | 0.9 | 1.3 | 1.7 |
| | PF | 0.0 | 0.1 | 0.1 | 0.1 |
| | S | 0.0 | 0.0 | 0.1 | 0.1 |
| Colta | Pr | 9.3 | 8.1 | 7.2 | 6.0 |
| | C | 0.1 | 0.5 | 0.7 | 0.9 |
| | Pz | 0.2 | 0.9 | 1.6 | 1.2 |
| | PF | 0.0 | 0.0 | 0.1 | 0.1 |
| | S | 0.1 | 0.1 | 0.2 | 0.2 |
| Guamote | Pr | 25.8 | 23.7 | 22.1 | 19.0 |
| | C | 0.3 | 0.7 | 1.3 | 1.7 |
| | Pz | 1.5 | 3.0 | 4.0 | 6.0 |
| | PF | 0.1 | 0.1 | 0.2 | 0.2 |
| | S | 0.1 | 0.2 | 0.3 | 0.3 |
| Guano | Pr | 3.3 | 2.6 | 2.2 | 2.0 |
| | C | 0.1 | 0.2 | 0.3 | 0.5 |
| | Pz | 0.2 | 0.7 | 0.9 | 2.1 |
| | PF | 0.0 | 0.0 | 0.0 | 0.1 |
| | S | 0.0 | 0.0 | 0.1 | 0.2 |
| Penipe | Pr | 7.3 | 6.1 | 5.2 | 3.0 |
| | C | 0.1 | 0.1 | 0.1 | 0.1 |
| | Pz | 0.3 | 1.1 | 1.8 | 2.0 |
| | PF | 0.1 | 0.1 | 0.1 | 0.1 |
| | S | 0.1 | 0.3 | 0.6 | 1.0 |
| Riobamba | Pr | 37.9 | 34.9 | 30.7 | 28.0 |
| | C | 0.9 | 1.4 | 2.6 | 4.4 |
| | Pz | 3.0 | 5.1 | 7.9 | 9.5 |
| | PF | 0.2 | 0.3 | 0.4 | 0.3 |
| | S | 0.1 | 0.3 | 0.5 | 0.6 |
| | | 100 | 100 | 100 | 100 |

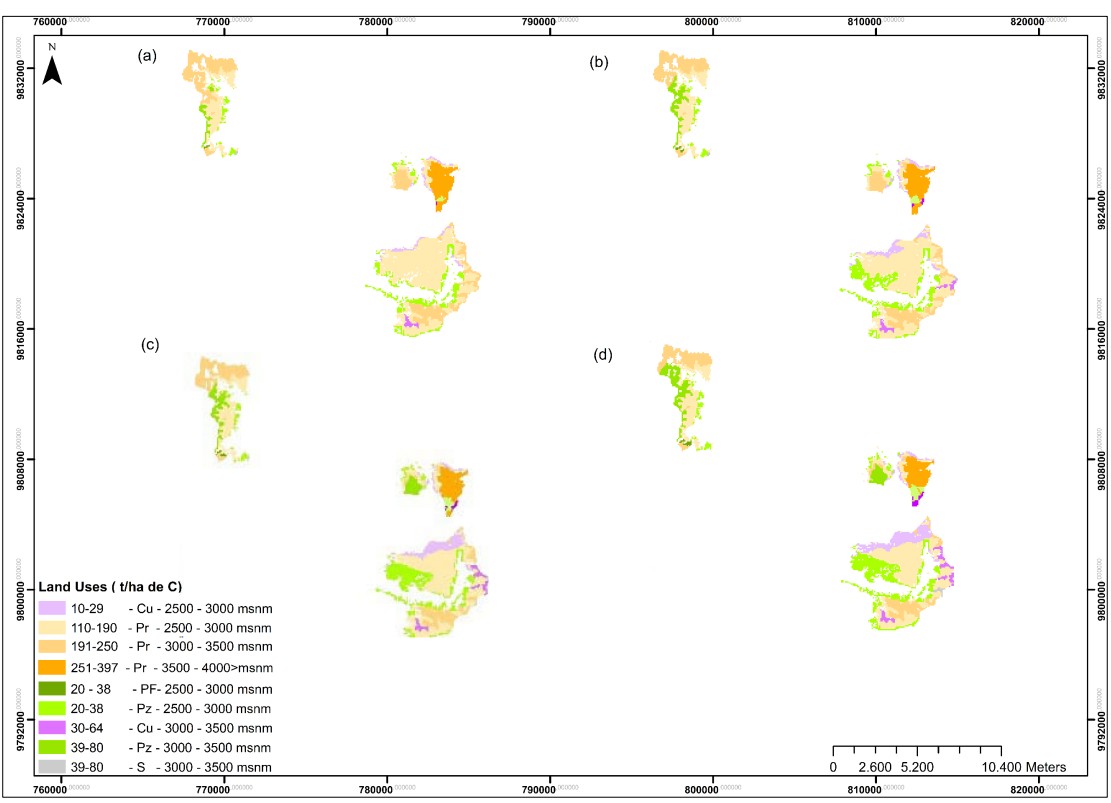

**Figure 8.** Land Uses—(**a**) Riobamba 2000; (**b**) Riobamba 2010; (**c**) Riobamba 2020; (**d**) Riobamba 2030.

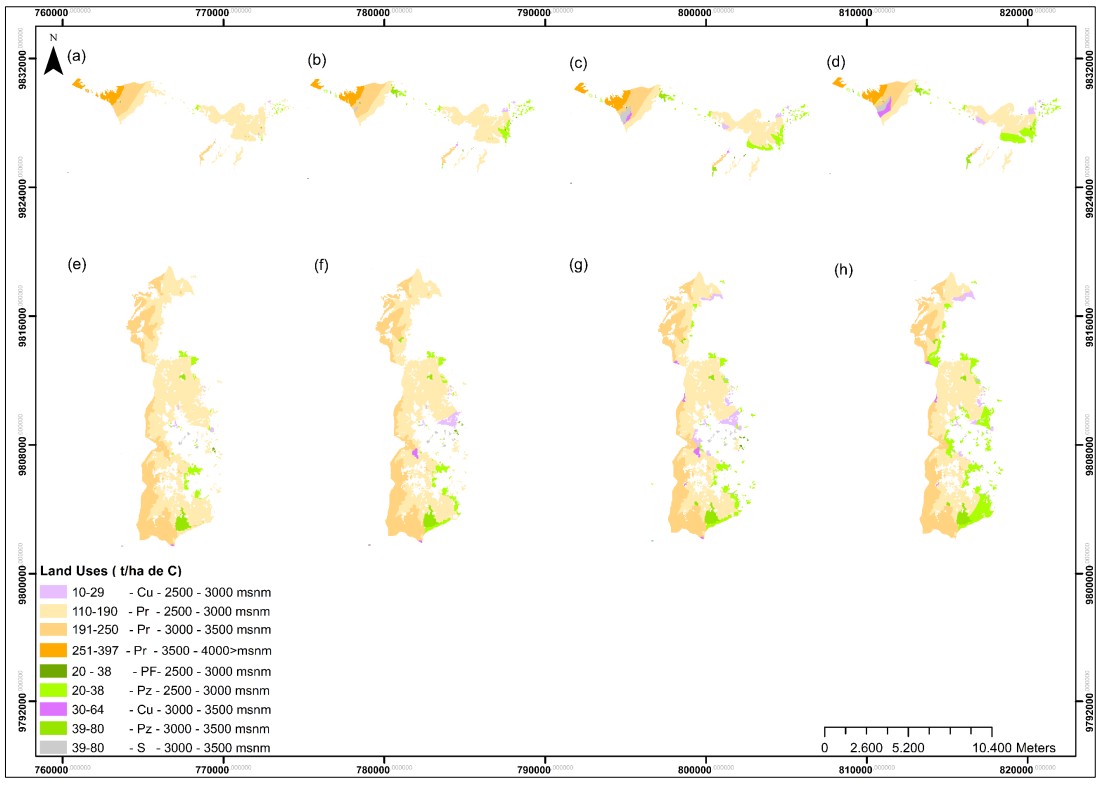

**Figure 9.** Land Uses—(**a**) Guano 2000; (**b**) Guano 2010; (**c**) Guano 2020; (**d**) Guano 2030; (**e**) Colta 2000; (**f**) Colta 2010; (**g**) Colta 2020; (**h**) Colta 2030.

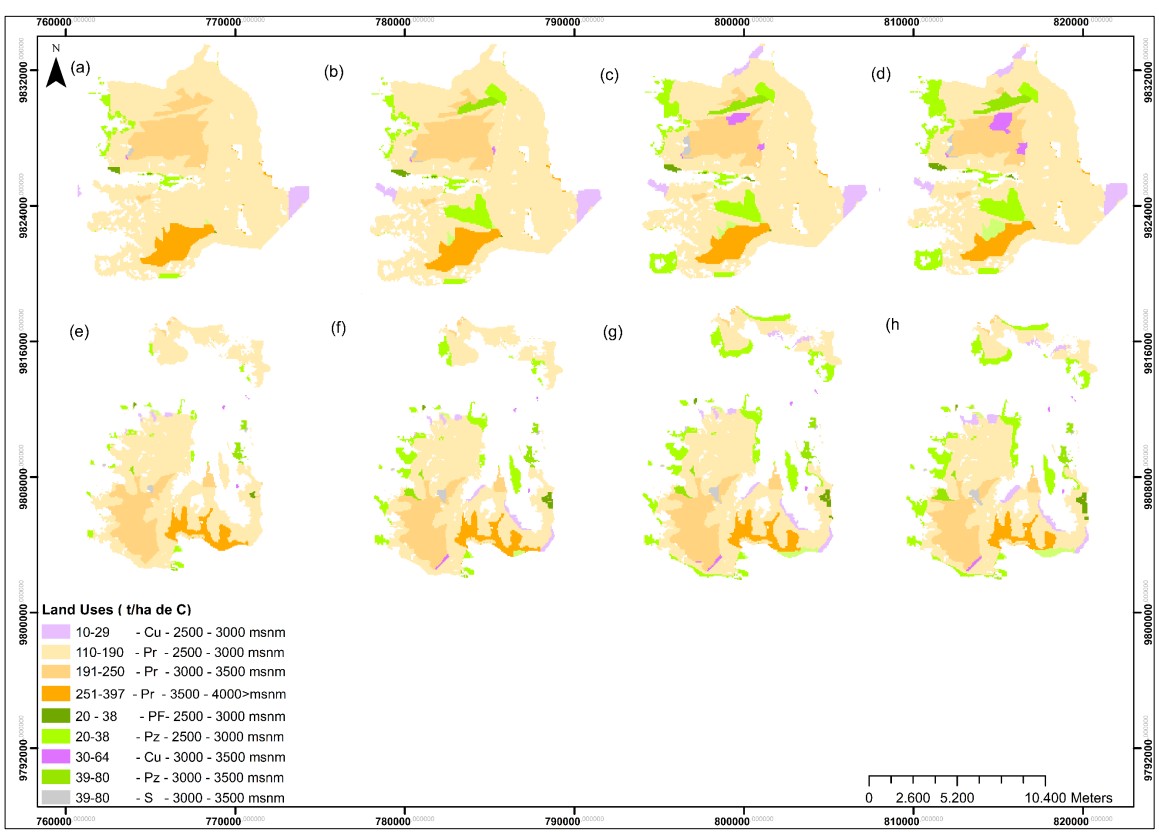

**Figure 10.** Land Uses—(**a**) Chambo 2000; (**b**) Chambo 2010; (**c**) Chambo 2020; (**d**) Chambo 2030; (**e**) Penipe 2000; (**f**) Penipe 2010; (**g**) Penipe 2020; (**h**) Penipe 2030.

**Table 7.** Gains, losses, and persistence of land uses in the Chambo sub-basin 2000–2010.

| Land Uses | 2000 | | 2000–2010 | | 2010 |
|---|---|---|---|---|---|
| | % | Persistence % | Gain% | Loss % | % |
| Pr | 92.1 | 79.5 | 3.5 | 12.6 | 83.0 |
| C | 1.6 | 0.6 | 3.0 | 1.0 | 3.6 |
| Pz | 5.6 | 5.0 | 7.0 | 0.6 | 12.0 |
| PF | 0.4 | 0.1 | 0.6 | 0.3 | 0.7 |
| S | 0.3 | 0.1 | 0.6 | 0.3 | 0.7 |
| | 100 | | | | 100 |

**Table 8.** Gains, losses and persistence of land uses in the Chambo sub-basin 2010–2020.

| Land Uses | 2010 | | 2010–2020 | | 2020 |
|---|---|---|---|---|---|
| | % | Persistence % | Gain % | Loss % | % |
| Pr | 83.0 | 63.9 | 10.1 | 19.1 | 74.0 |
| C | 3.6 | 1.0 | 5.3 | 2.6 | 6.3 |
| Gr | 12.0 | 7.0 | 10.8 | 5.0 | 17.9 |
| PF | 0.7 | 0.2 | 0.7 | 0.5 | 0.9 |
| S | 0.7 | 0.4 | 0.6 | 0.3 | 1.0 |
| | 100 | | | | 100 |

**Table 9.** Gains, losses, and persistence of land uses in the Chambo sub-basin 2020–2030.

| Land Uses | 2020 | 2020–2030 | | | 2030 |
| --- | --- | --- | --- | --- | --- |
| | % | Persistence % | Gain % | Loss % | % |
| Pr | 74.0 | 50.0 | 13.6 | 24.0 | 63.6 |
| C | 6.3 | 3.2 | 7.8 | 3.1 | 11.0 |
| Pz | 17.9 | 10.3 | 13.0 | 7.6 | 23.3 |
| PF | 0.9 | 0.2 | 0.9 | 0.8 | 1.0 |
| S | 1.0 | 0.4 | 0.8 | 0.6 | 1.2 |
| | 100 | | | | 100 |

The transitions of the páramo soils are directly related to the economic activities in the area. Although there is recovery of the páramo soils, the loss of the ecosystem is notable. The degradation of the resource will increase in the future as long as the degradation trend continues. Increasingly, the degradation of the páramo ecosystem compared to the total area of the sub-basin is as follows: Alausí 1.1%, Guano 1.2%, Chambo 2.0%, Colta 3.2%, Penipe 4.3%, Guamote 6.8%, and Riobamba 9.6%.

Riobamba and Guamote are the most degraded areas because they are located on the western edge of the Chambo sub-basin. Entry is easier due to their altitude, which has made it possible for production areas to grow. Forests and soil without vegetation have an important presence in these two sites. The use of the land for pasture is the main cause of the degradation of the páramo, probably due to the price stability policies of the dairy industry [21].

Crops also represent one of the lands uses that has caused the greatest environmental impact on the resource, perhaps because the soils in the study area have the appropriate characteristics for the development of tubers. Tubers are one of the main foodstuffs consumed by the inhabitants of the area and of the country; therefore, many farmers base the exploitation of their land on this type of crop [1]. The bare soils become more visible in the period studied, the increase in coverage could be attributed to the abrupt changes in temperature characteristic of the place, these variations cause dryness in the soil, causing loss of vegetation.

The dynamics of land use revealed the following. The pastures in 2000, 2010, and 2020 occupied 5.6%, 12.0%, and 17.9%, respectively, of the overall area of the sub-basin; it is estimated that by 2030, pasture areas will occupy 23.3% of the sub-basin. Pasture is the cover that has had the greatest impact on the natural resource; its gain from 2000 to 2010 was 7.0% and from 2010 to 2020 it was 10.8%; based on the future projection made in the study, its gain will increase by 13.0% by the year 2030.

Crop cover in 2000 was 1.6% of the study area, in 2010 it was 3.6%, in 2020 it was 6.3% and by 2030 it will occupy 10.9% of the area. Crops have increased their land cover from 2000 to 2010 by 3.0% and from 2010 to 2020 by 5.2%; by 2030 their area will increase to 7.7%. Crops are replaced by pasture and the future projection is that they will continue to be replaced by this category. The loss of cropland from 2000 to 2010 was 0.9%, from 2010 to 2020 it was 2.6% and from 2020 to 2030 it is estimated to be 3.1%. The persistence of this type of land use within the study period will vary from 0.6% to 3.1%.

Forest plantations occupied 0.3%, 0.6%, and 0.9% in 2000, 2010, and 2020, respectively; by 2030 they will occupy 1.0%. They will decrease from 2000 to 2030 by 1.4% and their coverage will increase by 2.1%, so it is believed that there will not be a considerable variation in this coverage in the future. The land uses by which the moorlands have been replaced in greater proportion are pastures. According to the changes in land cover, the persistence of forest soils is established in a range of 0.1% to 0.2% from 2000 to 2030.

In 2000, bare soil was present in 0.3%, in 2010 it occupied 0.7%, in 2020 its area increased to 0.9% and it is estimated that by 2030 its coverage will increase to 1.1%. Its change could be

attributed to the increase in temperature in the area, caused by the release of organic carbon into the atmosphere, as a result of the productive activities of the agro-perquarian activities.

Páramo coverage suffered a loss of 12.6% of its surface area from 2000 to 2010. From 2010 to 2020, 19.1% of the resource was degraded and it is estimated that its loss will increase to 23.9% by 2030. According to the analysis, it was determined that in 2000 the páramo occupied 92.1% of the sub-basin, in 2010 82.9%, in 2020 73.9%, and based on the land use prediction developed in the study, it is estimated that by 2030 the resource will occupy 63.5% of the area. The most important transitions of the resource are represented by pastures and crops; however, the change of the resource to bare soils is also considerable, an effect that could be caused by a disorderly agricultural and livestock planning or by the increase in temperatures and droughts in the sub-basin.

### 3.3. Ecosystem Valuation

Tables 10 and 11 detail the ecosystem value per hectare calculated for the different land uses evaluated.

**Table 10.** Ecosystem value in relation to opportunity costs.

| Ecosystem Valuation—Opportunity Costs | | |
|---|---|---|
| **Año** | **C ($/ha)** | **Pz ($/ha)** |
| 2000 | 550 | 1000 |
| 2010 | 608 | 1313 |
| 2020 | 629 | 1370 |
| 2030 | 738 | 1401 |

**Table 11.** Ecosystem value in relation to the transfer of benefits.

| Ecosystem Valuation—Profit Transfer | | |
|---|---|---|
| **Pr ($/ha)** | **PF ($/ha)** | **S ($/ha)** |
| 2355 | 160 | 81 |

The predicted ecosystem values for each category were calculated using the ecosystem value (EV) coefficients adjusted for the land uses of the most representative crop in the area and pasture land use in relation to the dairy sector, the most important activity in the area. Table 10 refers to the variability of economic costs over time in relation to parameters such as: land preparation, planting, cultural work, harvesting, and yield. Monetary values increase due to the increase in investment costs directly linked to the improvement of planting technology, which is reflected in soil yields.

Table 11 details the average values determined from several studies collected in the research "The value of ecosystem services and the world's natural capital" [26]. The values of the coverages from the aforementioned study were selected taking into consideration: geological and biological characteristics of the resources, environmental benefits, and bibliographic relevance.

Mapping and Spatial Quantification of Ecosystem Valuation

Figure 11 shows the spatial transitions of the ecosystem value (EV) associated with the study area and Figures 12 and 13 show the changes, restoration, and degradation zones of the EV of the Chambo sub-basin during the study period.

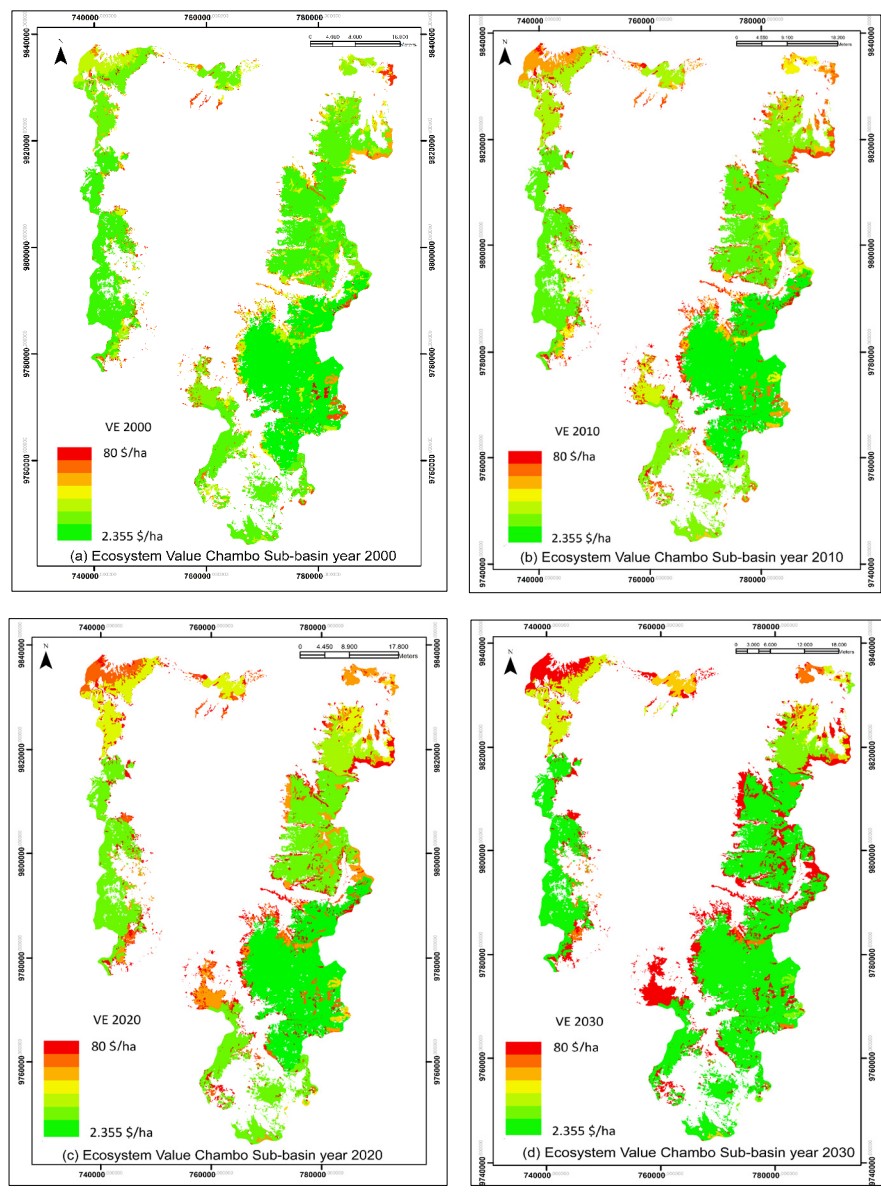

**Figure 11.** Ecosystem valuation of the Chambo Sub-basin (2000–2010–2020–2030).

The dynamics and trends of the EV must be understood simultaneously with the behavior of the páramo land use cover, since its variability is an effect produced directly by the changes in the cover of the resource.

The EV was calculated during the periods 2000–2010, 2010–2020 and a future projection for 2030 was considered; each period had a 10-year time span, so the economic values calculated refer to this time span.

The results show that each land use type had different changes in EV, as well as various gain and loss trends over the course of the study. For example, the EV of páramo from 2000, 2010, and 2020 was $2.78 \times 10^8$, $2.50 \times 10^8$, and $2.23 \times 10^8$ respectively it is estimated that by 2030 its value will decrease to $1.97 \times 10^8$.

From 2000 to 2020 pastures have gone from a value of $7.18 \times 10^6$ to $3.13 \times 10^7$, crops from $1.12 \times 10^6$ to $5.05 \times 10^6$, forest plantations from $7.52 \times 10^4$ to $1.77 \times 10^5$ and bare soils from $3.18 \times 10^4$ to $9.98 \times 10^4$ by 2030 it is estimated that pastures will take a value of $4.26 \times 10^7$, crops a value of $7.29 \times 10^6$, forest plantations a value of $2.23 \times 10^5$, and soils a value of $1.23 \times 10^5$.

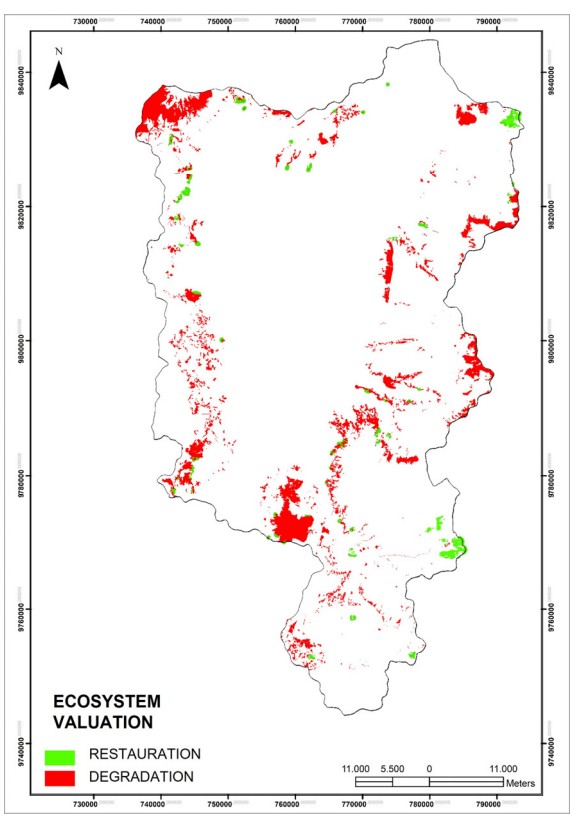

**Figure 12.** Restoration and degradation zoning of the EV.

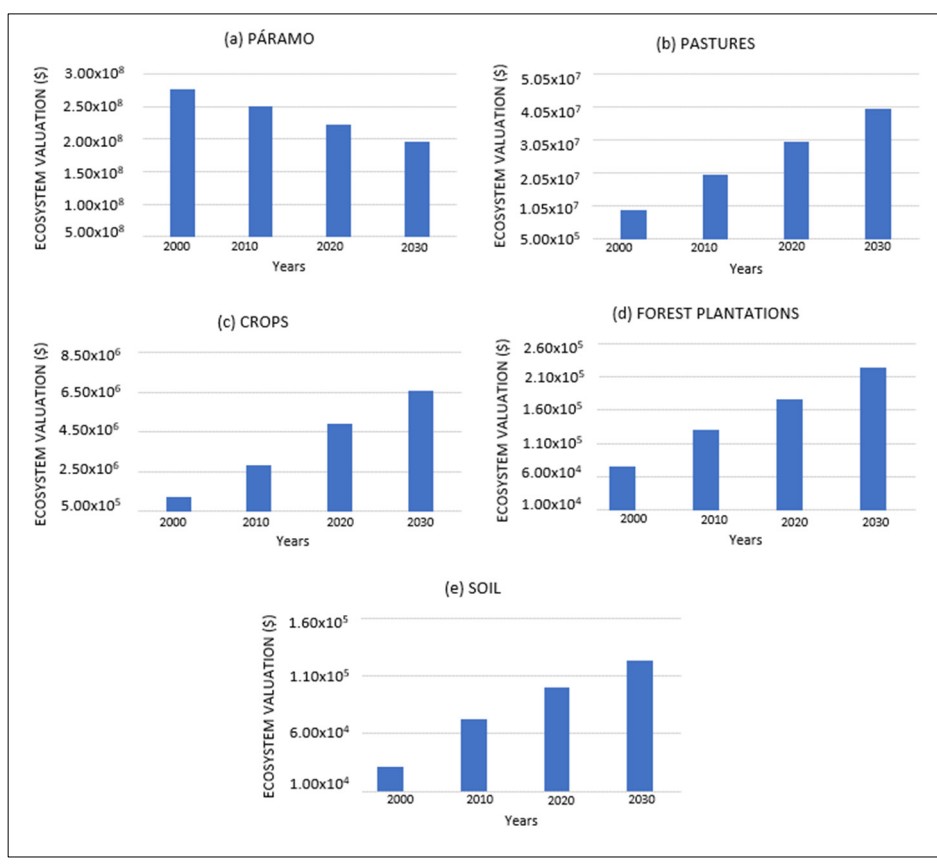

**Figure 13.** Variability of ecosystem valuation.

The VE of the site is linked to the SE of the páramo, so if this value increases or decreases, the availability of SE will be proportional. It was observed that at an altitude between 3500–4000 m.a.s.l, the Andean ecosystem accumulates from 252 to 359 t/ha of carbon, pasture cover has from 82 to 166 t/ha of carbon and crop categories have from 66 to 81 t/ha of carbon. In an altitudinal range between 3000 and 3500 m.a.s.l., the páramo has a carbon storage of 192 to 251 t/ha of carbon; crops conserve from 31 to 65 t/ha of carbon, pasture areas and soils without vegetation retain from 40 to 81 t/ha of carbon. Between 2500 to 3000 m.a.s.l, organic carbon rates ranged from 111 to 191 t/ha of carbon for the páramo ecosystem, 21 to 39 t/ha of carbon for the pasture category, and 11 to 30 t/ha of carbon for the cropland areas.

Carbon concentrations decreased as anthropogenic activities increased at different altitudinal levels, thus carbon release to the atmosphere increased and EV decreased.

Degradation of the EV in the area is more accentuated at altitudes of 2500 to 3000 m.a.s.l and 3000 to 3500 m.a.s.l, although the advance of the agricultural frontier is increasing, reaching the summit line in some sectors.

Riobamba was the canton in which the greatest degradation of the páramo's EV was detected, going from having a páramo area of $1.14 \times 10^8$ ha with a storage of 107 to 397 t/ha of carbon and an EV of $\$4.85 \times 10^4$ in 2010 to having a páramo area of $4.47 \times 10^4$ ha with an EV of $\$1.05 \times 10^8$ for the year 2020. It is estimated that in 2030 the EV will decrease to $\$8.27 \times 10^7$.

Páramo restoration areas are more accentuated in the areas adjacent to Sangay National Park. This type of change may occur as an effect of the conservation strategies adopted by the socio-páramo project, which pays an economic value to the owners of the Andean soils in compensation for letting the soils rest for a period. In the Guano canton, restoration areas can also be noted, which may be a consequence of the management plans adopted by its leaders, which are based on the fact that the municipality of the canton buys the most affected areas of land and protects them from any type of exploitation [21].

## 4. Discussion

The evaluation of land use changes related to their ecosystem valuation (EV) in areas that are not easily accessible due to climatic and geographic conditions can be supported by remote sensing methods and geographic information systems. For example, some authors [6,7] stated that the products generated through the detection of land use in conjunction with economic variables related to the benefits of environmental services can be the basis for developing methodologies that allow for understanding changes in the EV of natural resources [8].

The loss of ecosystem valuation (EV) of the study area from 2000 to 2020 was $\$5.50 \times 10^7$. This compares with other findings in southeastern and southwestern Ecuador, where the loss of EV was $\$5.45 \times 10^7$ and $\$5.40 \times 10^7$ [51,52], respectively in the same period.

The values of decrease in EV are close to those obtained in the present study. This may be due to the similarities in the areas analyzed, similar economic activities and policies focused on the protection of natural areas [49]. In another research conducted in northern Ecuador, the decrease in EV from 2000 to 2020 was $\$1.40 \times 10^7$ [53]. In comparison with the results of the (VE) obtained in the present work, the values are distant. The difference in these values may be associated with the variability of the environmental impacts produced on natural resources by economic activities and the capacity of environmental services (ES) of natural resources [54].

What has been stated by different authors regarding the variability of the EV of a natural environment from its land cover is aligned with the findings of this study, since it has been determined that Andean soils can be analyzed from an edaphic and economic approach by means of methods developed from satellite images and geographic information (GIS). These tools provide effective results, although it is important to mention that the accuracy of their results increases considerably when control points collected in the field are included.

For the case study, we determined that the NDVI, DEM, VARI, GSOC, BSI, NDMI, and control points obtained in the field were sufficient variables for the detection of páramo land uses and that their results allowed us to determine the land cover transitions, which were linked to the economic indexes adjusted for the study area. From the information provided by the methodology developed in the study, it was possible to recognize that factors such as colonization, population, and economic growth previously cited [1,2] are the main causes of soil changes. Páramo degradation can be attributed to practices with little or no conservation vision related to agriculture, cattle ranching, and reforestation, such as burning native vegetation to increase soil productivity levels [1].

The changes in the VE are stable throughout the period studied and related to land cover uses; the transitions are closely linked to the productive processes of the citizens [1]. This linkage contributes to understand that in the study area, both natural and social processes are related, so moorlands should be understood from a socio-ecological approach [2]. The loss of the EV not only affects soil productivity but also causes alterations in carbon concentrations, loss of native vegetation and degradation of resource properties [1].

The conservation strategies of Andean soils led to changes in land use, which implies that certain social sectors renounce income from agricultural production, so understanding the relationship between the economic rent of anthropogenic uses and natural capital is very important [44]. The monetary values obtained in relation to agricultural land use in this study have proven to be a useful spatial indicator that provides basic information for calculating the costs associated with natural resource management; these costs reflect the heterogeneity of land use in the natural environment [44].

Information determined from land use change assessment and ecosystem valuation can be applied to Andean landscape planning from an ex-ante and ex-post approach. Ex-ante when the spatial layers of land cover and EV are used in targeting analyses of vulnerable zones and better conserved areas. Ex-post when the information is used as a key performance indicator of economic budgets used in resource conservation strategies [49].

Spatial layers of land use and EVs can be used as a basis for cost-effectiveness audits of public investment in Andean conservation and justification of future programs. Incorporation of spatial indicators in the planning of priority areas to restore the natural capital of the páramos would improve the effectiveness of conservation plans and prioritize factors of social, economic, and environmental interest, allowing to adjust to limited budgets for the management of Andean resources. [50].

The results of this study highlight that, based on its methodology, a source of historical data on the variability of land use and EV has been generated. It differs from other studies by developing a method with specific thresholds for the recognition of Andean soils, taking as a study site the least studied area of the country. Based on the methods developed in the research, it will be possible to understand the spatiotemporal variability of land use and EV of the páramo in the past, present and future from a monetary approach.

The products to reproduce the research are affordable. By means of the functions defined in the study, the replication of the methods will be fast. The products generated can be used to begin to study the Ecuadorian páramo, from a monetary approach without the need to obtain large economic support to carry out this type of evaluations.

The information gathered in the study allows us to strategically focus the areas to develop conservation and restoration processes for the natural landscapes of the Andes Mountains, since the methodology can be extrapolated. Based on the evaluation of the conservation status of the páramo, it is necessary to implement timely conservation strategies in accordance with the reality of the geographic area studied. The updates of land cover and EV data sources are currently insufficient. It is essential to develop conservation plans for the Andean areas, otherwise the degradation of the ecosystem will lead to significant alterations in ecosystem services, for example, inconvenient water supply to the highlands region.

*Limitations of the Study*

The methods used have generated good results and have met the objectives set out in this study; however, it is important to describe some limitations of the methodology so that future researchers can consider them. The main limitations are as follows:

Number of variables: more spectral and environmental variables could be included to improve the effectiveness of the model; in this work it was complicated to do so due to the lack of information; for a long time, the Andean zones were little valued and studied.

Satellite images: attention should be paid to the calibration, atmospheric, and topographic correction of the images.

Learning methods: this method can relate variables as appropriate because it does not have a fixed distribution, so it can sometimes be a problem to choose an appropriate final answer. To confirm the effectiveness of the method, a confirmation of the data should be made from the control points collected in the field.

Ecosystem valuation: the information obtained in the study related to the EV of the area could be subjective as there may be changes among individuals and societies that are part of the economic activities over time, which would change the condition of the variables and therefore the results [47].

The methodology assumes that the monetary benefits from the exploitation of the natural resource will be increasing, and it is possible that this will change due to the viability of the country's economic development.

Errors may occur in EV mapping with respect to land use due to the nature of the data used in the analyses. For example, the spatial distribution of production costs may vary depending on farmers' land use techniques, crop rotations, weather changes, and soil properties. Uncertainty can also be generated by the variability of product yields over time [49].

The information obtained in relation to EV can be interpreted in different ways, as it will depend on the theoretical approach from which the analysis is conducted, which may influence the statements about the study area.

## 5. Conclusions

The proposed methodology to evaluate spatiotemporal changes in land use and EV of the páramo has proven to be adequate for the purpose of this work. It was determined that temporal monitoring of vegetation cover is essential to recognize the areas of greatest vulnerability to EV degradation. The thresholds, functions and interrelationships determined in the Multilayer Perception (MLP), Markov Chains (MC) and Automata Cells (AC) models for land cover recognition were good; the models had a performance of 80%. The hybridization of the MC and AC models is appropriate for predicting future land cover changes in the Andean zones in relation to their EV, the methods achieved an acceptable accuracy to recognize the land uses of the páramo. The variables chosen for land cover classification were adequate to identify the particularities of the land covers.

The mapping generated allowed us to determine that the páramo decreased by 13% between 2000–2010 and 19% between 2010–2020. It was estimated that the loss of the ecosystem from 2000 to 2030 will increase to 28%. From the first year of the study to the last year considered in the work, the páramo will go from occupying 92% to 64% of the area studied.

The predicted results of the EV revealed that the categories with anthropogenic activity analyzed maintain a constant growth that directly impacts the EV of the páramo. The most affected areas are those up to 3500 m.a.s.l. The EV of the Chambo sub-basin from 2000 to 2020 will go from $2.86 \times 10^8$ to $2.59 \times 10^8$ and it is estimated that by 2030 the EV will decrease to $2.48 \times 10^8$, which leads us to recognize that although the loss of the EV of the natural resource is not critical, its degradation is on the rise.

The ES decrease their capacity as the EV of the páramo degrades. Soil production capacity decreased throughout the study period, mainly affecting the cantons of Riobamba

and Guamote. The concentration of carbon and EV was higher at higher altitudinal levels, perhaps because of the climate and topography at that altitude.

Economic indicators from a spatial approach related to páramo cover are useful for quantifying opportunity costs and undertaking management actions, such as ecological restoration and sustainable planning of Andean landscapes.

**Author Contributions:** Conceptualization, Y.C.P., M.V. and J.J.d.F.; methodology, Y.C.P., J.J.d.F., M.V., F.C. and Y.P.; software, Y.C.P., M.V. and F.C.; validation, Y.C.P., M.V. and J.J.d.F.; formal analysis, Y.C.P., M.V., J.J.d.F., Y.P. and F.C; investigation, Y.C.P., F.C. and M.V.; resources, Y.C.P., Y.P., F.C. and M.V.; data curation, Y.C.P., Y.P. and J.J.d.F.; writing—original draft preparation, Y.C.P. and J.J.d.F.; writing—review and editing, Y.C.P., J.J.d.F., M.V., F.C. and Y.P.; visualization, Y.C.P. and Y.P.; supervision, Y.C.P., Y.P. and F.C; project administration, J.J.d.F. and Y.C.P. All authors have read and agreed to the published version of the manuscript.

**Funding:** This research received no external funding.

**Institutional Review Board Statement:** Not applicable.

**Informed Consent Statement:** Not applicable.

**Data Availability Statement:** The data presented in this study are available in article.

**Acknowledgments:** We thank Juan Jorge Sánchez for his academic support in this research and the Sustainability PhD Program of the Polytechnic University of Catalonia.

**Conflicts of Interest:** The authors declare no conflict of interest.

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
