# Peer review of "Evaluation of the Synergies of Land Use Changes and the Quality of Ecosystem Services in the Andean Zone of Central Ecuador"

_applsci, doi:10.3390/app14020498_

Round 1

Reviewer 1 Report

Comments and Suggestions for Authors

It is a routine study, without innovation. And the method is too simple, some important information is missed.

What is the scientific problem of this study, what is the innovation, please strength this point in the introduction section.

In the classification of land use, the threshold was set 30% cloud cover, which is too high, how to guarantee the classification quality?

The following set of variables were used to model the land uses of the Chambo sub-basin páramo: Normalized Difference Vegetation Index (NDVI), height (DEM), Green Atmosphere Resistant Visible Vegetation Index (VARI), Bare Soil Index (BSI), normalized difference moisture index (NDMI) and soil organic carbon (SOC). What the sources of these datasets. Or how to calculate them? Please introduce.

In line 175, the abbv. MLP can be used directly.

Fig.4-12, the legend is too small, it is not clear.

Author Response

Dear reviewer

The authors thank you for your comments, which were very valuable and constructive for the improvement of the work. Below, we detail the improvements made and answer the questions raised.

  1. It is a routine study, without innovation. And the method is too simple, some important. information is missed. What is the scientific problem of this study, what is the innovation, please strength this point in the introduction section.

The information supporting the potential and contributions of the study was included in the manuscript in the introduction and discussion section.

It is important to mention that the study should not be evaluated as an isolated remote sensing study, since the central focus of the work is to develop a methodology to understand the changes in páramo land use from an economic perspective by means of remote sensing techniques, geographic information systems and economic indices.

The problem of the study was described. This refers to the lack of studies that evaluate land use changes from an environmental and economic perspective. In many cases, until we start talking in monetary terms, the necessary importance is not given to natural resource conservation plans. It is necessary to know the changes in land use, but leaving the knowledge to that point limits the understanding of the value of the resource from an economic perspective that includes the social, economic and environmental sectors. This study seeks to go further and understand past, present and future changes in páramo land use through its ecosystem value, i.e. the economic value of páramo land use, linked to its natural value over time. Currently, the evaluation of the ecosystemic valuation of natural resources is a novel topic, in which there is still a long way to go.

For the analysis of páramo land use, this study used the artificial neural network of multilayer perception (MLP), which has gained recognition for its good performance. Although it is a reliable prediction model, its performance is often diminished by errors in the weight adjustments of the modeler's variables. This work will put efforts in decreasing as much as possible this type of errors, adjusting the weights through a comparison of the training values, the network output values and the expected values. The error will be reduced by comparing the interactions of the weights in the output signals, through the feedbacks of the network, based on the adjustment of the functions used in the synaptic links of the neurons.

To complement the analysis of land use changes and evaluate transitions of the ecosystem value of the resource in monetary terms, this research used the methodology of opportunity costs and benefit transfer following the pioneering work of Costanza, who estimated coefficients to determine the ecosystem value of natural biomes. The present work will focus on adjusting the coefficients determined by Costanza so that they are applicable to the Andean zones of Ecuador, by means of economic indexes.

In conclusion, this work made the following contributions:

  • Developed a methodology that allows a bioeconomic evaluation of the páramo ecosystem through time with a future projection, from a natural approach that includes economic factors.
  • Optimized the performance of the artificial neural network of multilayer perception (MLP), adjusting the weights of the variables in the synaptic links of the algorithm to improve the classification of land uses in the study area.
  • Performed the calculation of coefficient adjustment, using economic indices to analyze the ecosystem valuation of environmental services of páramo land uses. By means of these calculations, the monetary indexes used in the work reflect the reality of the study area and solve the disadvantage of other studies that used general coefficients to determine the economic values of natural systems. By adjusting the coefficients, the ecosystem valuation becomes real and is no longer subjective.

The advancement of ecosystem valuation methodologies is important since many of them have been criticized for their subjectivity. This study seeks to contribute to this line of research in order to generate information that will help establish payment strategies for ecosystem services as economic instruments that will help adopt land use practices that guarantee the provision of ecosystem services to the community with a vision of the future and open spaces to understand the importance of the páramo in environmental and economic terms.

  1. In the classification of land use, the threshold was set 30% cloud cover, which is too high, how to guarantee the classification quality?

The atmospheric correction procedure was detailed in the manuscript. In addition, it is important to mention that the manuscript describes that the cloudiness threshold was less than 30%, not that the threshold is 30%. The satellite images covered part of the Sierra and Amazon regions. The study area is located in the Andes mountain range, that is, in the center of the country, in the Sierra region. The highest percentage of clouds in the images was found in the Amazon region. The images were checked to ensure that the geographic space where the study area was located was free of clouds, and a cut was made to the images to specifically process the study area. A set of 20 satellite images was used and 3580 control points were established, 70% of which were verified in situ to ensure classification. Atmospheric correction was performed through QGIS software by means of the dark object subtraction method, which aims to correct the electromagnetic energy dispersion effect of water particles suspended in the atmosphere, thus subtracting from the image the values they add to the scene data.

  1. The following set of variables were used to model the land uses of the Chambo sub-basin páramo: Normalized Difference Vegetation Index (NDVI), height (DEM), Green Atmosphere Resistant Visible Vegetation Index (VARI), Bare Soil Index (BSI), normalized difference moisture index (NDMI) and soil organic carbon (SOC). What the sources of these datasets. Or how to calculate them? Please introduce.

A description of data sources, formulas and software of the variables used was included in the manuscript.

  1. In line 175, the abbv. MLP can be used directly.

The abv. MLP was used directly. In the first paragraph of the Land Use Classification Algorithms section.

  1. 4-12, the legend is too small, it is not clear.

Figures were redistributed and improved. The size of the captions was increased.

Additional

The introduction was improved by including important references that highlight the background and relevance of the work performed. The research design and methods were described in greater detail, diagrams and formulas were added. In the results section, a discussion was included that compares the data obtained in other investigations with the information provided in the work. Through this discussion, data that provide reliability to the results were provided. The conclusions were modified by highlighting the data obtained from the main objective of the research. The main objective of the research is to develop a methodology to determine the ecosystemic value of páramo soils by means of remote sensing techniques, geographic information systems and economic indexes.

The quality of writing, spelling and language editing has been improved with the support of native English speakers who are highly qualified in editing research literature.

All changes made are highlighted in the manuscript.

Reviewer 2 Report

Comments and Suggestions for Authors

1. Abstract: The significance of this study needs further elaboration to better communicate its importance and contributions.

2. Introduction: Highlight the key research points and avoid unclear statistical technique descriptions. Explicitly outline the progress in land-use prediction models for clarity.

3. References: Ensure software references include a website link and access date.

4. Section 2: Clarify the structure. Separate the remote sensing and data preprocessing explanation from the Methods section for better organization.

5. Line 223: Correct 'ReLu' to 'ReLU'.

6. Research Workflow: Clarify the focus and innovations, possibly using an overall research workflow diagram to improve clarity.

7. Methods Comparison: Highlight the importance of specific methods used by comparisons with other models. Include a comparison between the model used in this study and other models to establish reliability. Explain the MLR technique for predicting land-use changes more clearly within the Methods section.

8. Human Factors: Elaborate on human-induced factors influencing land-use changes in more detail.

9. Figure Improvements: Enhance figure clarity by adjusting font sizes and ensuring consistency. Use sub-picture descriptions for multiple framed images, e.g., Figure 1, and consider including distribution maps for factors like NDMI, SOC, etc.

The revision needs to address these points thoroughly. The manuscript has potential but requires significant refinement and clarification to meet the publication standards.

Author Response

Dear reviewer

The authors thank you for your comments, which were very valuable and constructive for the improvement of the work. Below, we detail the improvements made and answer the questions raised.

  1. Abstract: The significance of this study needs further elaboration to better communicate its importance and contributions.

At the beginning of the summary section, a couple of lines explaining the main reason for the study were included. The scarcity of information that allows understanding the importance of natural resources from an economic approach is often a limiting factor in establishing parameters related to environmental investment in conservation plans. The importance of the study lies in this aspect.

It is important to mention that the study should not be evaluated as an isolated remote sensing study, since the central focus of the work is to develop a methodology that allows understanding the changes in the use of páramo land from an economic perspective by means of remote sensing techniques, geographic information systems and economic indexes. This type of methodologies will contribute to establish payment strategies for ecosystem services as economic instruments, which will help to adopt land use practices that guarantee the provision of ecosystem services to the community with a vision of the future and open spaces to understand the importance of the páramo in environmental and economic terms.

Due to the limited number of words requested in the abstract (200 words). The main results obtained with respect to the main objective of the study are explicitly detailed. The contributions of the work are described in greater detail in the introduction and discussion section.

  1. Introduction: Highlight the key research points and avoid unclear statistical technique descriptions. Explicitly outline the progress in land-use prediction models for clarity.

The introduction has been improved by adding new references related to the study. The following is a brief description of the new structure of the introduction.

SECTION 1: This section briefly described the characteristics of the study site, including climatological, topographical and environmental features. The ecological importance of the natural resource was highlighted.

SECTION 2: The main factors that have contributed to the degradation of the ecosystem value of páramo soils were detailed. Human-induced factors influencing land use changes were highlighted. In addition, the environmental impacts generated by the degradation of the páramo ecosystem were described.

SECTION 3: The problem of the study was described. This refers to the lack of studies that evaluate land use changes from an environmental and economic perspective. In many cases, until we start talking in monetary terms, the necessary importance is not given to natural resource conservation plans. It is necessary to know the changes in land use, but leaving the knowledge to that point limits the understanding of the value of the resource from an economic perspective that includes the social, economic and environmental sectors. This study seeks to go further and understand past, present and future changes in páramo land use through its ecosystem value, i.e. the economic value of páramo land use, linked to its natural value over time.

SECTION 4: The progress of land use prediction models is detailed, with a brief description of their processes and limitations.

SECTION 5: References are made to studies of ecosystem valuation of land uses carried out in Ecuador and abroad, briefly detailing their results and conclusions.

SECTION 6: The methodology for the evaluation of land use that will be used in this study is described, and the contribution that the work will make to the methodology is detailed. It also describes the ecosystem valuation methodology included in the study and its contribution.

SECTION 7: Finally, the purpose, contributions and importance of the study are detailed.

  1. References: Ensure software references include a website link and access date.

Se incluyó en el documento el enlace al sitio web y fechas de acceso de los softwares utilizados.

Referencias añadidas:

Quantum GIS (QGIS). Disponible en línea: https://qgis.org/es/site/forusers/download.html (consultado el 10 de abril de 2023).

TerrSet Geospatial Monitoring and Modeling Software. Disponible en línea: https://clarklabs.org/terrset/ (acceso el 10 de abril de 2023).

  1. Section 2: Clarify the structure. Separate the remote sensing and data preprocessing explanation from the Methods section for better organization.

The structure was clarified as recommended. The new structure is briefly explained below.

Section 1

  • Study Area: main characteristics of the study area
  • Workflow: synthesis of the work performed
  • Satellite Images: Characteristics of the imagery used
  • Image Processing: Atmospheric and geometric image corrections
  • Checkpoints: Number of training points and sources
  • Variables: Variable sources, formulas and descriptions
  • Land Use Classification Algorithms: explanation of the algorithms used, each stage is covered.
  • Future land use prediction model: prediction technique, software and validation.

Section 2:

  • Evaluation of land use change: Statistical measures used to validate the algorithms.

Section 3:

  • Estimation of the ecosystem value of the zone: Description of the methods used to associate land use change with the ecosystem value of the resource based on market values.
  1. Line 223: Correct 'ReLu' to 'ReLU'.

The document has been carefully revised and 'ReLu' has been changed to 'ReLU'.

  1. Research Workflow: Clarify the focus and innovations, possibly using an overall research workflow diagram to improve clarity.

A workflow diagram was added to the document in the materials and methods section. The research approach and innovations were highlighted in the introduction and results section.

  1. Methods Comparison: Highlight the importance of specific methods used by comparisons with other models. Include a comparison between the model used in this study and other models to establish reliability. Explain the MLP technique for predicting land-use changes more clearly within the Methods section.

A graphic with the general scheme of the MLP model was added for better understanding and the description of the algorithm structure was clarified. In addition, a section explaining the procedures for each of the layers to find the model fit was added, including equations and descriptions.  In the introduction, the importance of the methodology used was highlighted. The results covering the main objective of the study, i.e. the analysis of the variability of the ecosystem valuation of páramo soils, were compared with the results of previous studies in the discussion section.

  1. Human Factors: Elaborate on human-induced factors influencing land-use changes in more detail.

Information on human-induced factors influencing land use change has been detailed in the paper.  The following is a brief description of the information included in the paper.

Cultural adaptation is the process by which man makes effective use of the energy potential of the Andean zones for productive purposes. The factors induced by humans in the changes in the use of páramo soils are linked to the economic activities of the area. The main economic activities in the area are based on agriculture and livestock and the secondary activities are based on mining and forestry.

Most of the farmers in the area plant crops without any planning system that includes rest periods and spatial logistics of the soil, causing environmental alterations, especially to the flora, fauna and soil resources.

The inhabitants who are dedicated to livestock farming use straw burning systems to fertilize the soil to increase pasture productivity. In addition, in order to increase the productivity of the páramo, they plant artificial pastures, which require a greater water supply than the natural pasture in the area. Artificial pastures have less trampling capacity, causing long-term erosion problems.  In general, this production system has a high economic cost due to its low productivity and a high ecological cost due to the deterioration of the soil on the slopes.

At first glance, it would seem that silvicultural activity favors ecosystem stability by planting trees in the páramo. However, this approach is not so true, since planting trees involves removing soil and vegetation from the area, which causes ecosystem alterations. In addition, in many cases the planted trees absorb quantities of water that compromise the stability of the environment. 

Mining activity in the study area is carried out on an artisanal basis and although it is not on a large scale, it causes land instability, displacement of animals, and elimination of natural flora.

The main effects of the degradation of the moorlands due to the aforementioned activities are as follows:

  • Indiscriminate elimination of native forests.
  • Loss of the activity of soil microorganisms
  • Alteration of nutrient cycling
  • Leaching of minerals
  • Accelerated mineralization of organic matter
  • Alteration of plant succession processes
  • Loss of refuge, nesting and feeding sites for wild fauna.
  • Alteration of microclimates
  1. Figure Improvements: Enhance figure clarity by adjusting font sizes and ensuring consistency. Use sub-picture descriptions for multiple framed images, e.g., Figure 1, and consider including distribution maps for factors like NDMI, SOC, etc

The images have been adjusted and labels have been used for multiple images. Soil organic carbon (SOC) values are already included in the maps in Figures 5 and 6. For NDMI and other spectral indices used in the study, no maps are included; these variables were specifically used as predictors and the results obtained from them are already reflected in the maps of the land use figures.

Additional

The introduction was improved by including important references that highlight the background and relevance of the work performed. The research design and methods were described in greater detail, diagrams and formulas were added. In the results section, a discussion was included that compares the data obtained in other investigations with the information provided in the work. Through this discussion, data that provide reliability to the results were provided. The conclusions were modified by highlighting the data obtained from the main objective of the research. The main objective of the research is to develop a methodology to determine the ecosystemic value of páramo soils by means of remote sensing techniques, geographic information systems and economic indexes.

The quality of writing, spelling and language editing has been improved with the support of native English speakers who are highly qualified in editing research literature.

All changes made are highlighted in the manuscript.

Reviewer 3 Report

Comments and Suggestions for Authors

Comments for “Evaluation of the conservation status of the páramo ecosystem of central Ecuador: an analysis based on the detection of land 4 use changes and ecosystem value.” by Yadira Carmen Pazmiño et al. A bioeconomic monitoring was carried out using remote sensing techniques to evaluate the effects of changes over time on the economic value of páramo land use, linked to its natural value.

The work gives some new insights for the ecosystem management based on the páramo decreased. And in the same time, there are some comments as following, which might be useful.

1.       The title could be shorter and focused.

2.       Line 20-21, “MLP”, “CM-AC” and “NDMI” should be given the full name when they were used first time. Pls check carefully all the text.

3.       Line 32, paramos or páramos? Should be uniformed.

4.       Line 38, change “Seventy-five percent” to “75%”.

5.       Line 127, Figure 1, the right part could be moved and merged with the left sub-figure. There is enough space.

6.       Line 240, the uncertainty of the prediction model should be discussed further more.

7.       Figure 4-12, it would be better to merge these figures to one. The text (including the numbers) is too small and be difficult to read.

8.       It would be better to merge some short paragraphs. There are too much paragraphs with only one sentence.

Comments on the Quality of English Language

The English is ok to read, but need mind the spelling such as the abbreviations.

Author Response

Dear reviewer

The authors thank you for your comments, which were very valuable and constructive for the improvement of the work. Below, we detail the improvements made and answer the questions raised.

  1. The title could be shorter and focused.

The title was modified highlighting the objective of the study in a concrete way.

New title:

Evaluation of the quality of ecosystem services and land use change of the páramo of central Ecuador.

  1. Line 20-21, “MLP”, “CM-AC” and “NDMI” should be given the full name when they were used first time. Pls check carefully all the text.

The lines, 20-21, "MLP", "CM-AC" and "NDMI" have been given full names and the entire text has been carefully revised.

  1. Line 32, paramos or páramos? Should be uniformed.

The text has been carefully revised and the paragraphs have been left in a uniform manner.

  1. Line 38, change “Seventy-five percent” to “75%”.

The change "seventy-five percent" to "75%" has been made.

  1. Line 127, Figure 1, the right part could be moved and merged with the left sub-figure. There is enough space.

The images in figure 1 were moved and merge

  1. Line 240, the uncertainty of the prediction model should be discussed further more.

A more in-depth discussion of the factors analyzed in the model uncertainty has been made in the article.  A brief explanation is detailed below.

The uncertainty of the prediction model was determined by considering: data quality, model validation and a sensitivity analysis. The data used were thoroughly reviewed to be representative of the real world being modeled, biases in the data that may significantly affect the reliability of the model were eliminated.

Validation of the prediction model was determined by the total number of control points included by the operator for each of the topographic coverages in the learning and validation process.  The performance trend was reflected in the classification of categories from the total number of objects recognized by the program. The degree of uncertainty will reflect the values observed in the image of the analyzed coverages and the values estimated by the classifier.

Sensitivity analysis made it possible to evaluate the robustness of the Markov chain model. The analysis is based on systematically varying the input parameters or assumptions of the model and observing the resulting changes in the predictions, the sensitivity of the model to different scenarios can be evaluated. This analysis helped to identify critical factors that significantly influence model predictions and provided information on the reliability of the model under various conditions.

Through the study of data quality, model validation and sensitivity analysis, model performance and uncertainty were established. The statistical measures used in these processes are shown in Table 1.

  1. Figure 4-12, it would be better to merge these figures to one. The text (including the numbers) is too small and be difficult to read.

The images were merged into two figures. The amount of information did not allow merging the images into a single figure. The resolution of the figures was improved.

  1. It would be better to merge some short paragraphs. There are too much paragraphs with only one sentence.

The entire document was reviewed and as many paragraphs as possible were merged without losing the context.

Adicional

The introduction was improved by including important references that highlight the background and relevance of the work performed. The research design and methods were described in greater detail, diagrams and formulas were added. In the results section, a discussion was included that compares the data obtained in other investigations with the information provided in the work. Through this discussion, data that provide reliability to the results were provided. The conclusions were modified by highlighting the data obtained from the main objective of the research. The main objective of the research is to develop a methodology to determine the ecosystemic value of páramo soils by means of remote sensing techniques, geographic information systems and economic indexes.

The quality of writing, spelling and language editing has been improved with the support of native English speakers who are highly qualified in editing research literature.

All changes made are highlighted in the manuscript.

Reviewer 4 Report

Comments and Suggestions for Authors

Dear authors,

The manuscript needs some improvements before it can be published. Please improve the manuscript according to the review. Please pay more attention to the purpose of the study.

Best

Comments on the Quality of English Language

-Some sentences should be improved

-The flow of the sentences/paragraphs should be improved.

Author Response

Dear reviewer

The authors thank you for your comments, they were very valuable and constructive for the improvement of the work. Below, we detail the improvements made.

  1. Please present one or two sentences explaining the reason/s why this study should be conducted.

At the beginning of the summary section, a couple of lines explaining the main reason for the study were included. The scarcity of information that allows understanding the importance of natural resources from an economic approach. On many occasions, not being able to value natural systems based on monetary values represents a limitation to establish parameters related to environmental investment in conservation plans. The importance of the study lies in this aspect.

It is important to mention that the study should not be evaluated as an isolated remote sensing study, since the central focus of the work is to develop a methodology that allows understanding the changes in land use in páramo from an economic perspective by means of remote sensing techniques, geographic information systems and economic indexes. This type of methodologies will contribute to establish payment strategies for ecosystem services as economic instruments, which will help to adopt land use practices that guarantee the provision of ecosystem services to the community with a vision of the future and open spaces to understand the importance of the páramo in environmental and economic terms.

Due to the limited number of words requested in the abstract (200 words). The main results obtained with respect to the main objective of the study are explicitly detailed. The contributions of the work are described in greater detail in the introduction and discussion section.

  1. What are the limitations of the previous studies, please explain, then what is/are the novelty/novelties of your current research?. This sentence may not complete enough, the purpose of the study only because of the least studied area of Ecuador's paramo?.

The introduction section was modified. Previous studies were described including their limitations. In addition, an explanation of the importance and new contributions made by the current research was made.

In the final part of the introduction, the purpose of the research was complemented by the objectives.

  1. Please do not write a paragraph only consists of one sentence, so just integrate this sentence with its following pargraph. It is better to write a paragraph consists of more than two sentences and no more than 6 sentences. Please do it for the whole manuscript.

The entire document has been revised and all one-sentence paragraphs of the manuscript have been integrated.

  1. Please present the equations used in this study.

All equations used in this study have been included. Including those used to extract spectral indices.

  1. Cifras poco claras, especialmente la legenda.

The legends of all the figures in the work have been clarified.

  1. Please compare the EV of the current study with the previous studies.

A comparison of the ecosystem valuation results of the current work with the results of previous studies has been included.

Additional

The introduction was improved by including important references that highlight the background and relevance of the work performed. The research design and methods were described in greater detail, diagrams and formulas were added. In the results section, a discussion was included that compares the data obtained in other investigations with the information provided in the work. Through this discussion, data that provide reliability to the results were provided. The conclusions were modified by highlighting the data obtained from the main objective of the research. The main objective of the research is to develop a methodology to determine the ecosystemic value of páramo soils by means of remote sensing techniques, geographic information systems and economic indexes.

The quality of writing, spelling and language editing has been improved with the support of native English speakers who are highly qualified in editing research literature.

All changes made are highlighted in the manuscript.

Reviewer 5 Report

Comments and Suggestions for Authors

Dear authors, you can find my suggestions/criticisms and contributions regarding the article in the attached file.
Best wishes

Comments on the Quality of English Language

-

Author Response

Dear reviewer

The authors thank you for your comments, they were very valuable and constructive for the improvement of the work. Below, we detail the improvements made according to what was recommended.

  • The whole document has been revised and the m.a.s.l. terms have been left.
  • The suggested lines of text have been eliminated.
  • The section detailing the backpropagation algorithm of the MLP model was described in more detail.
  • Altitude has been left as a variable
  • The images were improved
  • Recommended reference information has been included
  • The sentence that was in Spanish has been corrected and translated into English.

Additional

The introduction was improved by including important references that highlight the background and relevance of the work performed. The research design and methods were described in greater detail, diagrams and formulas were added. In the results section, a discussion was included that compares the data obtained in other investigations with the information provided in the work. Through this discussion, data that provide reliability to the results were provided. The conclusions were modified by highlighting the data obtained from the main objective of the research. The main objective of the research is to develop a methodology to determine the ecosystemic value of páramo soils by means of remote sensing techniques, geographic information systems and economic indexes.

The quality of writing, spelling and language editing has been improved with the support of native English speakers who are highly qualified in editing research literature.

All changes made are highlighted in the manuscript.

Round 2

Reviewer 1 Report

Comments and Suggestions for Authors

I have no more comments.

Author Response

Dear reviewer
The authors thank you for your comments, they were very valuable and constructive for the improvement of the work. 

Kind regards
The authors

Reviewer 2 Report

Comments and Suggestions for Authors

1. Introduction: It's recommended to refine and streamline the introduction for better focus and clarity, ensuring a concise yet comprehensive overview.

2. Figures: The subgraphs within Figures 7 and 8 appear too small. Considering the limited space, each figure should represent one year, while the additional data could be provided as supplementary material. For Figures 9 and 11, subfigure numbers should be employed to enhance their visual impact and clarity.

These issues can be effectively addressed with a minor revision. The manuscript demonstrates promising insights into páramo land use variations and EV within the Chambo-Ecuador sub-basin. Further enhancements will contribute to the paper's overall quality.

Author Response

Dear reviewer

The authors thank you for your comments, which are very valuable and constructive for the improvement of the work. The following are the improvements that have been made.

  1.  Introduction: It's recommended to refine and streamline the introduction for better focus and clarity, ensuring a concise yet comprehensive overview.

The introduction was simplified, ensuring a concise description.

  1. Figures: The subgraphs within Figures 7 and 8 appear too small. Considering the limited space, each figure should represent one year, while the additional data could be provided as supplementary material. For Figures 9 and 11, subfigure numbers should be employed to enhance their visual impact and clarity.

Figures 7 and 8 were re-distributed. Two figures were added to increase the size of the land use maps. As complementary material, one map is sent for each canton of the study area.

The maps were not grouped by year because by joining the cantons to delimit the sub-basin, sensitivity to observe land use changes clearly was lost.

The subfigures in Figures 9 and 11 were labeled to improve their visual impact and clarity.

Adicional

New references have been removed and added.

Best regards
